# LLM Watermark Evasion via Bias Inversion

**Jeongyeon Hwang** [1]  **Sangdon Park** [1]  **Jungseul Ok** [1]

## Abstract

Watermarking offers a promising solution for detecting LLM-generated content, yet its robustness under realistic query-free (black-box) evasion remains an open challenge. Existing query-free attacks often achieve limited success or severely distort semantic meaning. We bridge this gap by theoretically analyzing rewriting-based evasion, demonstrating that reducing the average conditional probability of sampling green tokens by a small margin causes the detection probability to decay exponentially. Guided by this insight, we propose the *Bias-Inversion Rewriting Attack* (BIRA), a practical query-free method that applies a negative logit bias to a proxy suppression set identified via token surprisal. Empirically, BIRA achieves state-of-the-art evasion rates ($> 99\%$) across diverse watermarking schemes while preserving semantic fidelity substantially better than prior baselines. Our findings reveal a fundamental vulnerability in current watermarking methods and highlight the need for rigorous stress tests. Our code is available at here.

## 1. Introduction

The rapid advancement and proliferation of large language models (LLMs) (Minaee et al., 2024; Wang et al., 2024a) have intensified concerns about misuse, ranging from the spread of misleading content (Monteith et al., 2024; Wang et al., 2024b; Papageorgiou et al., 2024) to threats to academic integrity such as cheating (Stokel-Walker, 2022; Kamalov et al., 2023). To address these risks, watermarking has been proposed as a promising approach for detecting LLM-generated content (Aaronson & Kirchner, 2022; Kirchenbauer et al., 2024b). The core idea is to embed an imperceptible statistical signal into generated text, for example, by partitioning the vocabulary into "green" and "red" lists using a secret key and biasing generation toward the green list. A detector then flags LLM-generated text by testing for a statistical overrepresentation of green tokens.

Prior works (Kirchenbauer et al., 2024b; Liu et al., 2024; Zhao et al., 2024; Lu et al., 2024) suggest that watermarking can withstand common post-edit operations such as insertion, substitution, and deletion, fueling interest in real-world deployment (Bartz & Hu, 2023; Tong, 2024).

At the same time, recent studies (Raffel et al., 2020; Cheng et al., 2025; Wu & Chandrasekaran, 2024; Chen et al., 2024; Jovanović et al., 2024; Diaa et al., 2024) question whether current watermarking methods are sufficiently stress-tested. Existing evasion approaches broadly fall into two categories. *Query-based* attacks (Wu & Chandrasekaran, 2024; Chen et al., 2024; Jovanović et al., 2024) use repeated queries to infer the green token set and then remove the signal. However, these methods require access to the watermarked model and partial knowledge of the watermarking scheme, making them impractical for text of unknown provenance. In contrast, *query-free* attacks (Krishna et al., 2023; Cheng et al., 2025) operate in a realistic black-box setting and usually rely on LLM-based rewriting. However, existing query-free methods achieve only limited evasion success and often substantially distort the original semantics.

In this paper, to address these limitations and deepen our understanding of watermarking vulnerabilities, we theoretically analyze rewriting attacks from first principles and show that it suffices for a rewriter (an LLM) to reduce the *average conditional probability* of sampling green tokens by a margin $\delta > 0$ to make the detection probability vanish *exponentially* in $\delta^2$. This characterization directly motivates a practical query-free attack, which we call the *Bias Inversion Rewriting Attack* (BIRA). BIRA leverages this theoretical insight during rewriting by applying a negative logit bias to a proxy suppression set identified via token self-information, thereby reducing the overrepresentation of green tokens.

Empirically, this simple intervention enables BIRA to evade recent watermarking schemes, even beyond the green-red setting, outperforming prior query-free attacks while better preserving semantics. These results suggest that many current watermarks can be removed with minimal effort, highlighting the need for more robust watermarking and evaluations beyond simple editing-based stress tests.

Our contributions are summarized as follows:

[1]Pohang University of Science and Technology (POSTECH), South Korea. Correspondence to: Jungseul Ok <jungseul@postech.ac.kr>.

*Proceedings of the 43rd International Conference on Machine Learning*, Seoul, South Korea. PMLR 306, 2026. Copyright 2026 by the author(s).

- We develop a theoretical analysis of watermark rewriting attacks, deriving a sufficient condition for successful evasion.

- Building on this insight, we propose BIRA, a simple and practical query-free attack that weakens the watermark signal by applying a negative logit bias to a proxy suppression set during rewriting.

- We conduct extensive experiments showing that BIRA achieves state-of-the-art evasion rates against recent watermarking methods while maintaining high semantic fidelity.

## 2. Related Work

**LLM watermarking.** Kirchenbauer et al. (2024a) introduced a widely used green–red list watermarking scheme that partitions the vocabulary into green and red subsets and embeds a detectable statistical signal by applying a positive logit bias to green tokens. Subsequent work has improved robustness by enhancing key generation and detection procedures (Kirchenbauer et al., 2024b; Liu et al., 2023a; Zhao et al., 2024; Liu et al., 2024; Lee et al., 2024; Lu et al., 2024) or by more faithfully preserving the original LLM output distribution (Wu et al., 2023). Other lines of research explore sampling-based watermarking approaches (Aaronson & Kirchner, 2022; Hu et al., 2023; Christ et al., 2024). More recently, sentence-level watermarking methods have been proposed to further enhance robustness by embedding signals at the semantic sentence level rather than the token level (Hou et al., 2024; Dabiriaghdam & Wang, 2025).

**Watermark evasion attacks.** Watermark evasion attacks can be broadly categorized under the threat model into two types: *query-based* and *query-free*. Query-based attacks (Jovanović et al., 2024; Chen et al., 2024; Wu & Chandrasekaran, 2024) infer the green-token set by issuing many crafted prefix prompts. While effective, they require both access to the watermarked model and knowledge of the watermarking scheme, making them impractical when either the model or the scheme is unknown.

By contrast, query-free attacks (Kirchenbauer et al., 2024a; Krishna et al., 2023; Cheng et al., 2025; Diaa et al., 2024) operate directly on generated text in a practical black-box setting, typically via LLM-based rewriting. Prior work either fine-tunes an LLM for paraphrasing (Krishna et al., 2023; Diaa et al., 2024) or masks and regenerates high-entropy tokens (Cheng et al., 2025). However, these methods often achieve limited evasion success and, crucially, can substantially distort meaning; without an explicit fidelity constraint, the task becomes degenerate, since one can simply regenerate fresh unwatermarked text with an unwatermarked LLM. In contrast, our method achieves markedly higher evasion rates while maintaining high semantic fidelity, without addi-tional training and under a black-box setup.

## 3. Preliminary

**Language model.** A language model, denoted by $\mathcal{M}$, generates text $y$ by predicting the next token in a sequence. Given an input sequence $x^{0:n-1} = [x^{(0)}, \ldots, x^{(n-1)}]$, the model outputs a logit vector $l^{(n)} = (l_0^{(n)}, \ldots, l_{V-1}^{(n)}) \in \mathbb{R}^V$ from which it derives a probability distribution $Q^{(n)}$ over the vocabulary $\mathcal{V}$ of size $V$ using the softmax operator:

$$Q_u^{(n)} = \frac{\exp(l_u^{(n)})}{\sum_{j=1}^{V} \exp(l_j^{(n)})}, \quad u \in \mathcal{V}.$$

The next token $x^{(n)}$ is then drawn from $Q^{(n)}$, either by sampling or by another decoding strategy.

**Watermarking algorithm.** A watermarking algorithm $\mathcal{W}$ consists of two components: a *generation function* $\mathcal{S}$ and a *detection function* $\mathcal{D}$. Given a secret key $k$, the algorithm $\mathcal{W}_k$ modifies the distribution $Q^{(n)}$ during text generation to produce $\widehat{Q}^{(n)} = \mathcal{M}(x^{0:n-1}, \mathcal{W}_k)$, embedding hidden patterns (e.g., green tokens) into the output $y$. For instance, Kirchenbauer et al. (2024a); Liu et al. (2023a); Zhao et al. (2024); Liu et al. (2024); Lee et al. (2024); Lu et al. (2024) add a positive logit bias $\gamma > 0$ to $l_u^{(n)}$ for tokens $u \in \mathcal{G}(\mathcal{W}_k)$, the green set generated by the secret key $k$, which increases their sampling probability and biases the generated text $\hat{y}$ toward green tokens. The detection function $\mathcal{D}$ then takes a text sequence $y$ and the same secret key $k$ as input, and determines whether $y$ is watermarked:

$$\mathcal{D}(y, \mathcal{W}_k) = \mathbf{1}\{Z(y; \mathcal{W}_k) \geq \tau\},$$

where $Z(y; \mathcal{W}_k)$ is a test statistic on the watermark patterns (e.g., a one-proportion $z$-statistic on the fraction of green tokens), and $\tau \in \mathbb{R}$ is the detection threshold. Here, the null hypothesis $H_0$ is that the text was not generated with $\mathcal{W}_k$, and the watermark is detected by rejecting $H_0$ when $Z(y; \mathcal{W}_k) \geq \tau$.

**Threat model.** We consider a black-box threat model where the adversary has no knowledge of the watermarking scheme $\mathcal{W}$ or the target model.

**Adversary's objective.** The adversary's goal is to design a text modification function $\mathcal{F}$ that transforms a watermarked text $\hat{y}$ into a modified text $\tilde{y} = \mathcal{F}(\hat{y})$, which is detected as unwatermarked, while preserving the original meaning of $\hat{y}$:

$$\mathcal{F}^* = \arg\min_{\mathcal{F}} \ \mathbb{E}\left[\mathcal{D}(\tilde{y}, \mathcal{W}_k)\right] \quad \text{s.t.} \quad \text{sim}(\tilde{y}, \hat{y}) \geq \epsilon, \quad (1)$$

where $\text{sim}$ is a similarity measure between two texts used to evaluate semantic preservation.

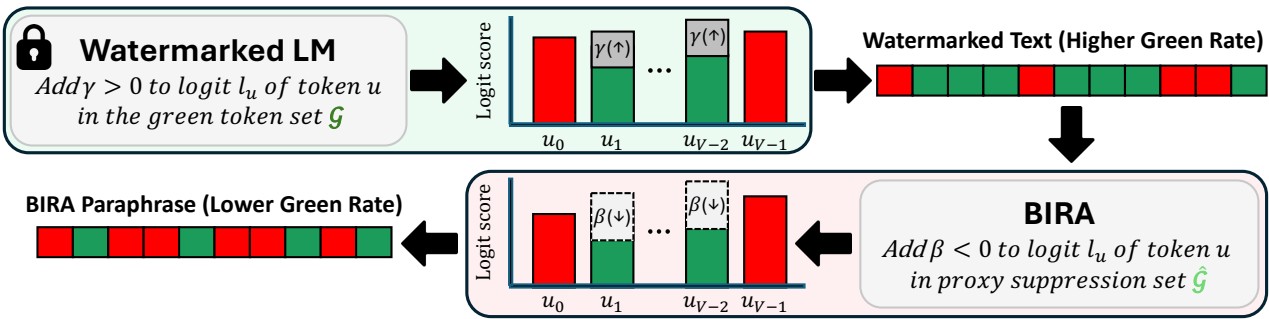

*Figure 1.* Illustration of BIRA. A watermarked LLM typically increases the likelihood of sampling green tokens by adding a positive bias $\gamma > 0$ to their logits at each generation step. In contrast, BIRA applies a negative bias $\beta < 0$ to a proxy suppression set, thereby suppressing their sampling probability. This inversion lowers the probability of generating green tokens and weakens the watermark signal, enabling the paraphrased text to evade detection.

## 4. Method

In this section, we first analyze rewriting-based evasion and derive sufficient conditions under which watermark detection fails (Section 4.1), and then translate this analysis into a practical query-free attack, *Bias Inversion Rewriting Attack* (BIRA) (Section 4.2). Figure 1 provides an overview.

### 4.1. Theoretical Analysis of Rewriting Attack

A watermark evasion attack aims to reduce the statistical overrepresentation of green tokens so that the resulting text no longer triggers the detector. We begin with a simple reduction: for any detector whose test statistic is a nondecreasing function of the empirical green-token rate, detection is equivalent to thresholding the empirical green-token rate $\hat{p}(y; \mathcal{W}_k)$ (Lemma 4.1). Building on this, we show that if an attack can keep the *average conditional* probability of sampling a green token below the induced threshold by a margin $\delta > 0$, then the detection probability decays exponentially in $\delta^2$ (Theorem 4.2).

**Lemma 4.1.** *Let the detector be* $\mathcal{D}(y, \mathcal{W}_k) = \mathbf{1}\{Z(y; \mathcal{W}_k) \geq \tau\}$ *and suppose there exists a nondecreasing function* $h : [0, 1] \rightarrow \mathbb{R}$ *with*

$$Z(y; \mathcal{W}_k) = h(\hat{p}(y; \mathcal{W}_k)),$$

$$\hat{p}(y; \mathcal{W}_k) = \frac{1}{N} \sum_{n=0}^{N-1} \mathbf{1}\{y^{(n)} \in \mathcal{G}(\mathcal{W}_k)\},$$

*where* $\mathcal{G}(\mathcal{W}_k)$ *denotes the green set produced by watermarking* $\mathcal{W}_k$. *Then, for a given* $N$, *there exists* $p_\tau \in [0, 1]$ *such that*

$$\mathcal{D}(y, \mathcal{W}_k) = \mathbf{1}\{\hat{p}(y; \mathcal{W}_k) \geq p_\tau\},$$

*with* $p_\tau = \inf\{p : h(p) \geq \tau\}$. *In particular, for the widely used one-proportion $z$-test for watermark detection, such a function $h$ exists.*

Lemma 4.1 shows that, for this class of detectors, watermark detection depends only on whether the empirical green-token rate $\hat{p}(y; \mathcal{W}_k)$ exceeds a threshold $p_\tau$. We next show

that if a rewriter (an LLM) can keep the *average conditional* probability of sampling a green token at least $\delta > 0$ below this threshold across the sequence, then the detection probability decays exponentially in $\delta^2$ (Theorem 4.2).

**Theorem 4.2.** *Let* $\tilde{y} = [\tilde{y}^{(0)}, \ldots, \tilde{y}^{(N-1)}]$ *be the rewriter's output, and let $p_\tau$ denote the induced green-rate threshold from Lemma 4.1. If there exists $\delta > 0$ such that*

$$\frac{1}{N} \sum_{n=0}^{N-1} \mathbb{E}\Big[\mathbf{1}\{\tilde{y}^{(n)} \in \mathcal{G}(\mathcal{W}_k)\} \,\Big|\, \tilde{y}^{0:n-1}\Big] \;\leq\; p_\tau - \delta,$$

*then*

$$\Pr\big[\mathcal{D}(\tilde{y}, \mathcal{W}_k) = 1\big] \;\leq\; \exp\Big(-\frac{1}{2} N \delta^2\Big).$$

Theorem 4.2 shows that if a rewriter can keep the *average conditional* probability of sampling a green token at least $\delta$ below the detector threshold $p_\tau$ across the sequence, then the detection probability decays exponentially in $\delta^2$. Thus, even a small per-step suppression of sampling the green-token probability, when achieved on average across the sequence, is sufficient to drive the overall detection probability toward zero. Proofs of Lemma 4.1 and Theorem 4.2 are provided in Appendix B.

**Application to KGW watermarking.** For KGW (Kirchenbauer et al., 2024a), the one-proportion $z$-statistic is $Z(y; W_k) = \big(\hat{p}(y; W_k) - p_0\big)/\sqrt{p_0(1-p_0)/N}$, where $p_0$ is the predefined green-token ratio and $N$ is the total number of generated tokens. Since $h(p) = \big(p - p_0\big)/\sqrt{p_0(1-p_0)/N}$ is nondecreasing, the threshold corresponds to $p_\tau = p_0 + \tau\sqrt{p_0(1-p_0)/N}$. For default setups $p_0 = 0.5$, $\tau = 4$, and $N = 230$, we obtain $p_\tau \approx 0.632$. If the attack keeps $\frac{1}{N}\mathbb{E}\big[\mathbf{1}\{\tilde{y}^{(n)} \in \mathcal{G}(W_k)\} \mid \tilde{y}^{0:n-1}\big] \leq 0.632 - \delta$, then by Theorem 2 the detection probability satisfies $\Pr[D(\tilde{y}, W_k) = 1] \leq \exp(-N\delta^2/2)$; e.g., $\delta = 0.1 \Rightarrow e^{-1.15} \approx 0.316$, $\delta = 0.2 \Rightarrow e^{-4.6} \approx 0.010$.

---

**Algorithm 1** BIRA (Bias-Inversion Rewriting Attack)

1: **Input:** System prompt $S$; Watermarked text $\hat{y}^{0:N-1}$; Language model $\mathcal{M}$; Percentile $q \in [0, 1)$; Initial bias $\beta_0 < 0$; lr $> 0$; Max restarts $R$; Max length $L_{\max}$; Window size $h$; threshold $\rho \in (0, 1]$.

2: *Phase 1: Construct Proxy Suppression Set $\widehat{\mathcal{G}}$*

3: Compute self-information $I^{(n)}$ for each token $\hat{y}^{(n)}$ using the language model $\mathcal{M}$.

4: **for** $n = 0$ **to** $N - 1$ **do**

5:     $I^{(n)} \leftarrow -\log P_{\mathcal{M}}(\hat{y}^{(n)} \mid \hat{y}^{0:n-1})$

6: **end for**

7: Set percentile threshold $\eta \leftarrow \text{Percentile}(\{I^{(n)}\}_{n=0}^{N-1}, q)$

8: Define the proxy suppression set $\widehat{\mathcal{G}} \leftarrow \{\text{id}(\hat{y}^{(n)}) \mid I^{(n)} \geq \eta, \; n \in [0, N-1]\}$

9: *Phase 2: Perform Bias-Inversion Rewriting*

10: $\beta \leftarrow \beta_0$

11: **for** $r = 1$ **to** $R$ **do**

12:     Initialize empty sequence $\tilde{y} \leftarrow []$

13:     **for** $t = 0$ **to** $L_{\max} - 1$ **do**

14:         Obtain logits $l^{(t)}$ from $M(\tilde{y}, \hat{y}^{0:N-1}, S)$

15:         Apply negative bias: $l_u^{(t)} \leftarrow l_u^{(t)} + \beta \cdot \mathbf{1}\{u \in \widehat{\mathcal{G}}\}$ for all $u$ in vocabulary

16:         Sample next token $\tilde{y}^{(t)} \sim \text{softmax}(l^{(t)})$

17:         Append $\tilde{y}^{(t)}$ to $\tilde{y}$

18:     **end for**

19:     Let $L$ be the length of $\tilde{y}$.

20:     **if** Distinct-1-Gram-Ratio$(\tilde{y}^{L-h:L-1}) < \rho$ **then**

21:         $\beta \leftarrow \min(0, \beta + \text{lr})$    *(Reduce strength of bias and restart)*

22:         **continue**

23:     **else**

24:         **return** $\tilde{y}$    *(Return text)*

25:     **end if**

26: **end for**

27: **return** $\tilde{y}$

---

## 4.2. Bias-Inversion Rewriting Attack

To translate Theorem 4.2 into a practical attack, one would ideally suppress tokens in the true green set $\mathcal{G}(\mathcal{W}_k)$, but $\mathcal{G}(\mathcal{W}_k)$ is inaccessible in a black-box setting. However, Theorem 4.2 implies that perfect identification is unnecessary: evasion only requires a small but consistent reduction in the probability of sampling green tokens. We therefore construct a global proxy suppression set $\widehat{\mathcal{G}}$ from the input text to target likely watermark traces during rewriting.

Although the true green list $\mathcal{G}(\mathcal{W}_k)$ is unknown, the watermarked text $\hat{y}$ contains the *realized* watermark trace: specific token choices fixed at generation time, and detection relies on these traces persisting under rewriting. In contrast, unwatermarked generation, produced without access to the secret key, does not systematically bias toward green tokens beyond the baseline rate. Motivated by this, we treat a subset

of tokens in $\hat{y}$ as a proxy for watermark traces and suppress their survival during rewriting to reduce green-token bias in the rewritten text.

A naive proxy is to include all input tokens in $\widehat{\mathcal{G}}$, but this can degrade text quality by suppressing frequent function words. Instead, we target tokens most likely to carry watermark influence. Prior work (Cheng et al., 2025) shows that watermarking schemes concentrate their effect on high-entropy positions to preserve utility, making high-surprisal tokens more likely to reflect watermark-biased choices. This is consistent with observations that watermarking is harder to embed in low-entropy settings such as code generation, where limited flexibility to bias token choice leads to fewer green tokens (Lee et al., 2024). We therefore construct $\widehat{\mathcal{G}}$ using token self-information, focusing suppression on likely traces while better preserving text quality.

Given a watermarked text $\hat{y} = [\hat{y}^{(0)}, \ldots, \hat{y}^{(N-1)}]$, we compute token surprisal under a public language model $\mathcal{M}$:

$$I^{(n)} = -\log P_{\mathcal{M}}\left(\hat{y}^{(n)} \mid \hat{y}^{(0:n-1)}\right).$$

Let $\eta$ be the $q$th percentile of $\{I^{(n)}\}_{n=0}^{N-1}$, and define

$$\widehat{\mathcal{G}} \leftarrow \left\{\text{id}(\hat{y}^{(n)}) \;\middle|\; I^{(n)} \geq \eta, \; n \in \{0, \ldots, N-1\}\right\},$$

where $\text{id}(\cdot)$ maps a token to its vocabulary index. During rewriting to produce $\tilde{y}$, we apply *bias inversion* by adding a negative logit bias $\beta < 0$ to tokens in $\widehat{\mathcal{G}}$ at every decoding step:

$$l_u^{(n)} \leftarrow l_u^{(n)} + \beta \, \mathbf{1}\{u \in \widehat{\mathcal{G}}\}, \qquad \forall u \in \mathcal{V}.$$

This discourages reproduction of watermark traces, and empirically, suppressing tokens in $\widehat{\mathcal{G}}$ suffices to reduce the *average* green-token sampling probability (Theorem 4.2).

### 4.2.1. MITIGATING TEXT DEGENERATION WITH ADAPTIVE BIAS

We observe that applying a strong negative bias $\beta$ can occasionally cause **text degeneration**, where the model repeatedly generates the same phrase (qualitative examples are provided in Appendix M.2). This arises from a distorted token distribution created by suppressing specific tokens and cannot be resolved by simply regenerating text, since the underlying distribution remains unchanged. To address this, our attack adaptively adjusts $\beta$ by detecting degeneration through monitoring the diversity of the last $h$ generated tokens. Specifically, we compute the distinct 1-gram ratio within this window, $\tilde{y}^{M-h:M-1}$, and classify the text as degenerated if the ratio falls below a predefined threshold $\rho$. The algorithms and details of degeneration detection are provided in Appendix C. Upon detection, the magnitude of negative bias is reduced for the next-generation attempt:

$$\beta \leftarrow \min(0, \beta + \text{lr}),$$

where $\mathrm{lr} > 0$ is a small step size. This adaptive adjustment allows the attack to begin with a strong bias for effective watermark removal and then gracefully reduce its strength only when necessary to prevent semantic degradation. The full procedure of our method is presented in Algorithm 1.

To initialize the logit bias $\beta$, we generate 50 paraphrases from the C4 dataset (Raffel et al., 2020) and gradually decrease $\beta$ (e.g., from $-1$ down to $-12$) until degeneration appears in at least one of the 50 outputs. We then use this value as the initial logit bias $\beta_0$, which strengthens the attack while minimizing the risk of degeneration. Since degeneration is rare (2.4% over 500 samples, with an average of only 1.03 iterations per text with $\mathrm{lr} = 0.125$), the computational overhead of the adaptive process is negligible.

# 5. Experiments

## 5.1. Setup

**Dataset.** For a fair comparison with SIRA (Cheng et al., 2025), the current state-of-the-art query-free attack, we follow its setup and generate watermarked text from C4 using the first 500 test samples as prompts. For each prompt, we generate 230 tokens with OPT-1.3B (Zhang et al., 2022) as the watermarked text generator. We observe that SIRA is effective only under this setup and fails to preserve semantics when the generation length is increased to 600 tokens, whereas our method remains effective (Section 5.3). We also evaluate on Dolly CW (Conover et al., 2023) and DBpedia (Raffel et al., 2020), where our method consistently outperforms the baselines (Appendix E.1).

**Watermark algorithms.** We evaluate seven watermarking methods: KGW (Kirchenbauer et al., 2024a), Unigram (Zhao et al., 2024), UPV (Liu et al., 2023a), EWD (Lu et al., 2024), DIP (Wu et al., 2023), SIR (Liu et al., 2024), and EXP (Aaronson & Kirchner, 2022). For each method, we use the default or recommended hyperparameters from the original studies (Pan et al., 2024; Cheng et al., 2025). For KGW, we use a single left hash on only the immediately preceding token to form the green and red token lists, since fewer preceding tokens improve robustness against watermark evasion attacks. Appendix E.2 further evaluates our method on recent sentence-level watermarking schemes (Dabiriaghdam & Wang, 2025; Huo et al., 2025), localized detectors (Kirchenbauer et al., 2024b), and failure modes that shift watermark signals to lower-entropy tokens or use stronger logit bias $\gamma$. Although our analysis focuses on token-level green-red watermarking, our method remains effective beyond this setting, and these failure modes show both impractical and ineffective.

**Baselines and language models.** We compare against three *query-free* attack baselines: Vanilla (paraphrasing with a language model), DIPPER (Krishna et al., 2023) (a trained paraphrasing expert), and SIRA (Cheng et al., 2025) (mask-ing and rewriting strategy). For DIPPER-1, we set lexical diversity to 60 without order diversity, and for DIPPER-2, we add order diversity of 40 to increase paraphrasing strength. We further compare against an adaptive attack (Diaa et al., 2024) that assumes knowledge of the watermarking scheme (Appendix H). Even in a black-box setup, our method outperforms this baseline with only a slight trade-off in semantic fidelity.

Across all methods, we use rewriting models (Llama-3.1-8B, Llama-3.1-70B, and GPT-4o-mini) that are independent of the watermark-generation model, and compute token surprisal with the same rewriting model. We use top-$p$ sampling ($p = 0.95$) with temperature 0.7; the rewriting prompt is provided in Appendix K. Since GPT-4o-mini does not expose logits, we use Llama-3.2-3B as an auxiliary model to estimate high-surprisal tokens, convert them to text, and re-tokenize them with the GPT tokenizer for SIRA and our method. We then apply negative logit bias through the GPT API, which supports token-level logit biasing. Appendix I shows that performance is robust to this tokenizer mismatch.

For SIRA, we set the masking threshold to $0.3$, as recommended by Cheng et al. (2025), and apply it to high-entropy tokens across all models. For our method, we use a percentile threshold of $q = 0.5$ to construct the proxy suppression set. The initial negative logit bias is $\beta_0 = -4$ for Llama-3.1-8B and Llama-3.1-70B (Dubey et al., 2023), and $\beta_0 = -11$ for GPT-4o-mini (OpenAI, 2024b), following the initialization strategy in Section 4.2.1, with $\mathrm{lr} = 0.125$.

### 5.1.1. EVALUATION METRICS

We evaluate attacks in terms of both attack efficacy and text quality.

**Attack efficacy.** Our primary measure is the *Attack Success Rate (ASR)*, the fraction of attacked watermarked texts that are misclassified as non-watermarked. To reduce sensitivity to detector threshold choices, following (Zhao et al., 2024; Liu et al., 2024; Cheng et al., 2025), we construct a test set of 500 attacked texts and 500 human-written texts and calibrate the detector's $z$-threshold to achieve False Positive Rates (FPR) of 1% and 10%. At these FPRs, we report the corresponding True Positive Rate (TPR) and F1-score. Lower TPR indicates that attacked texts are harder to distinguish from human-written text.

**Text quality.** We assess text quality using five metrics that cover semantic fidelity, paraphrasing strength, and fluency. To evaluate semantic preservation, we employ three measures. First, we use an **LLM judgement score** (Zheng et al., 2023; Fu et al., 2023; Liu et al., 2023b) from GPT-4o-2024-08-06 (OpenAI, 2024a), which scores meaning preservation on a 1-to-5 scale: a score of 5 indicates perfect fidelity, 4 allows for minor nuances without factual changes,

*Table 1.* Comparison of watermarking robustness under different attack methods. Our method, BIRA, achieves the highest attack success rate across all baselines.

| Attack \ Watermark | KGW | Unigram | UPV | EWD | DIP | SIR | EXP |
|---|---|---|---|---|---|---|---|
| Vanilla (Llama-3.1-8B) | 88.8% | 73.4% | 73.4% | 92.6% | 99.8% | 54.0% | 80.6% |
| Vanilla (Llama-3.1-70B) | 87.4% | 67.0% | 65.0% | 89.4% | 98.8% | 42.8% | 70.4% |
| Vanilla (GPT-4o-mini) | 60.2% | 30.2% | 46.8% | 58.8% | 95.8% | 23.6% | 31.8% |
| DIPPER-1 | 93.8% | 61.2% | 80.6% | 92.8% | 99.4% | 55.6% | 90.8% |
| DIPPER-2 | 97.2% | 71.8% | 85.4% | 96.6% | 99.2% | 70.4% | 97.2% |
| SIRA (Llama-3.1-8B) | 98.8% | 95.0% | 87.6% | 99.8% | 99.6% | 72.8% | 95.2% |
| SIRA (Llama-3.1-70B) | 98.0% | 87.6% | 85.0% | 99.2% | 99.6% | 60.6% | 88.6% |
| SIRA (GPT-4o-mini) | 98.0% | 85.2% | 84.8% | 97.2% | 99.6% | 57.6% | 94.8% |
| **BIRA (Llama-3.1-8B, ours)** | **99.8%** | **99.4%** | **99.8%** | **100.0%** | **100.0%** | **99.6%** | **99.8%** |
| **BIRA (Llama-3.1-70B, ours)** | **99.4%** | **99.0%** | **99.6%** | **99.8%** | **99.6%** | **98.8%** | **98.0%** |
| **BIRA (GPT-4o-mini, ours)** | **99.4%** | **100.0%** | **100.0%** | **99.8%** | **99.8%** | **99.4%** | **98.2%** |

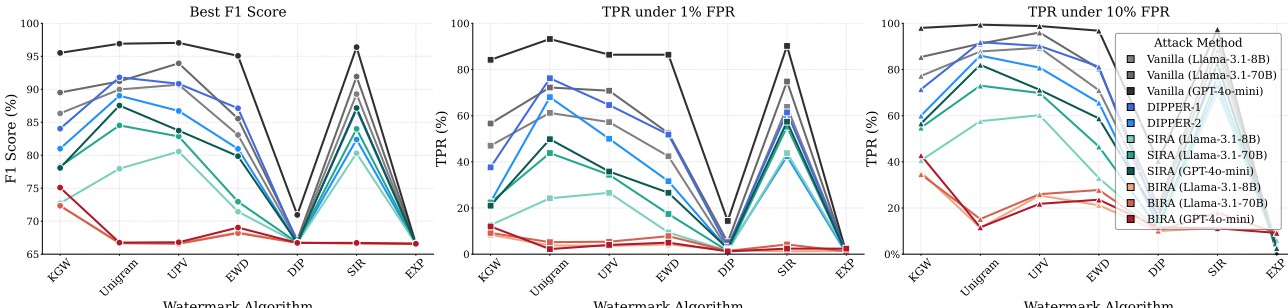

*Figure 2.* Comparison of detection performance with the adjusted threshold across watermarking algorithms, mitigating the effect of default threshold. We show the best F1 score (↓) and TPR (↓) at FPR of 1% and 10%. BIRA consistently achieves lower F1 and TPR than all baselines, indicating greater difficulty for detectors in distinguishing attacked text from human-written text. Exact values are provided in Appendix N.1.

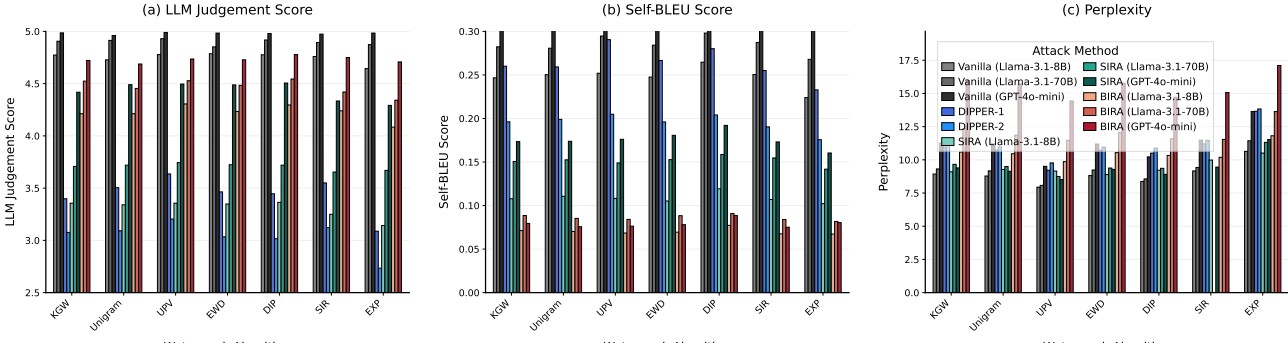

*Figure 3.* Comparison of text quality across different attacks for various watermarking methods, evaluated by LLM judgment score (↑), Self-BLEU score (↓), and Perplexity (↓). Our method preserves semantic fidelity to the original text compared to other attack baselines (DIPPER and SIRA) while providing stronger paraphrasing, as reflected in lower Self-BLEU scores. Additional results for NLI score (↑) and S-BERT score (↑) are provided in Figure 15 and exact values are detailed in Appendix N.2.

and 3 reflects that only the main idea is preserved while important details or relations are altered (see Appendix J for prompt details). We also compute an **NLI score** using `nli-deberta-v3-large` (He et al., 2020) to assess logical consistency between the original and attacked texts by evaluating mutual entailment. In addition, we report an **S-BERT score** (Reimers & Gurevych, 2019), following

(Cheng et al., 2025), which is based on the cosine similarity between sentence embeddings of the two texts.

To quantify the degree of paraphrasing and assess text naturalness, we use two additional metrics. Paraphrasing strength is measured with the **Self-BLEU score** (Zhu et al., 2018), which computes the BLEU score (Papineni et al., 2002) of each attacked text against its corresponding water-

**Watermarked Text by KGW ($z$-score: 6.03)**

Graciousness might not seem like the most important thing in defining the success of a nation, but it is paramount for Mr Lim Siong. By the age of 45, he already runs a fortune of close to $1bn. His wife, Ms Rachel Jia Xu (above), started a chain of popular supermarkets, which now has 13 outlets and employs more than 5,000 people. Not by sheer talent alone, but because of a strong sense of character that has made him a standout amongst the elite among Singapore's business leaders.

**Attacked Text by BIRA ($z$-score: 0.83)**

Graciousness may appear insignificant in determining what makes a country successful; however, it holds great importance for Mr. Lim Siong. At just 45 years old, he manages nearly $1 billion in wealth. His spouse, Ms. Rachel Jia Xu, has established an acclaimed supermarket franchise with currently 13 locations employing over 5,000 individuals. His prominence in Singapore's business community is attributed not solely to his exceptional skills but also his remarkable personal qualities.

*Figure 4.* Qualitative comparison of KGW-watermarked text and its BIRA rewrite (Llama-3.1-8B). BIRA suppresses green tokens while preserving meaning, reducing the $z$-score from 6.03 to 0.83. Additional examples (longer texts and other schemes) are in Appendix J.1.

*Table 2.* Effect of logit bias $\beta$ and percentile $q$ on attack performance

| Logit Bias ($\beta$) | 0.0 | -1.0 | -2.0 | -3.0 | -4.0 | -5.0 | -6.0 | -7.0 | -8.0 | -9.0 |
|---|---|---|---|---|---|---|---|---|---|---|
| ASR ($\uparrow$) | 54.2% | 74.4% | 88.2% | 97.2% | 99.6% | 99.4% | 99.6% | 99.8% | 99.8% | 99.6% |
| LLM Judgment ($\uparrow$) | 4.76 | 4.69 | 4.62 | 4.48 | 4.24 | 3.79 | 3.45 | 3.19 | 3.05 | 2.94 |
| Self-BLEU ($\downarrow$) | 0.25 | 0.20 | 0.15 | 0.11 | 0.07 | 0.04 | 0.03 | 0.02 | 0.01 | 0.01 |
| Perplexity ($\downarrow$) | 9.17 | 8.99 | 9.07 | 9.47 | 10.09 | 11.23 | 12.56 | 13.42 | 14.03 | 14.10 |
| Iteration ($\downarrow$) | 1.00 | 1.00 | 1.00 | 1.01 | 1.03 | 1.20 | 1.48 | 1.83 | 2.54 | 3.36 |
| Percentile ($q$) | 0.0 | 0.1 | 0.2 | 0.3 | 0.4 | 0.5 | 0.6 | 0.7 | 0.8 | 0.9 |
| ASR ($\uparrow$) | 99.0% | 99.6% | 99.2% | 98.6% | 99.6% | 99.6% | 98.4% | 96.4% | 89.0% | 77.2% |
| LLM Judgment ($\uparrow$) | 4.15 | 4.20 | 4.15 | 4.16 | 4.18 | 4.24 | 4.26 | 4.35 | 4.48 | 4.60 |
| Self-BLEU ($\downarrow$) | 0.06 | 0.06 | 0.06 | 0.06 | 0.06 | 0.07 | 0.08 | 0.10 | 0.13 | 0.18 |
| Perplexity ($\downarrow$) | 12.16 | 12.01 | 11.72 | 11.37 | 10.68 | 10.09 | 9.62 | 9.30 | 8.95 | 8.93 |
| Iteration ($\downarrow$) | 1.06 | 1.06 | 1.05 | 1.06 | 1.05 | 1.03 | 1.02 | 1.02 | 1.01 | 1.00 |

marked reference. This measures the overlap between the two texts, where a lower score indicates less lexical overlap and therefore stronger paraphrasing. Text naturalness is evaluated using **Perplexity (PPL)** (Jelinek et al., 1977), where a lower PPL corresponds to more probable and natural text.

### 5.2. Experimental Results

**Attack efficacy.** Table 1 compares watermark removal methods using ASR. BIRA consistently outperforms all baselines across watermarking algorithms and language models, with especially large gains against SIR, the strongest baseline watermark. For example, on GPT-4o-mini, vanilla paraphrasing achieves 49.6 ASR on average, whereas BIRA reaches 99.5. Additionally, Figure 2 reports detector performance under calibrated thresholds (FPR = 1% and 10%) and the best F1 score for each watermarking algorithm. Across settings, BIRA yields the lowest best F1 score and TPR, confirming its superior evasion effectiveness. Interestingly, BIRA is also effective against EXP, a sampling-based watermarking method, and sentence-level watermarks (Appendix E.2.1). For EXP, watermark traces are also concentrated in high-entropy token choices (Kirchenbauer et al., 2024b), which BIRA suppresses using self-information as

a proxy. For sentence-level watermarks, BIRA's document-level paraphrasing can merge or split sentences, disrupting the consistent sentence segmentation assumed by detection.

**Text quality.** Figure 3 shows that vanilla paraphrasing attains the highest LLM judgment score because its paraphrasing ability is weak and largely preserves the original structure, which leads to low ASR. This is consistent with its highest Self-BLEU score, indicating strong overlap with the source text. In contrast, our method achieves a significantly higher LLM judgment score than stronger baselines such as DIPPER and SIRA, demonstrating better semantic preservation. Notably, the state-of-the-art baseline SIRA often distorts meaning due to its masking procedure, which disrupts contextual information. At the same time, BIRA yields a much lower Self-BLEU score, showing that it generates more diverse paraphrases and relies less on reusing words from the watermarked text. For perplexity, our method remains comparable to other approaches, with only a slight increase when GPT-4o-mini is used. We attribute this increase to GPT-4o-mini occasionally generating stylistically stiff (e.g., excessive hyphenation) but semantically faithful text, leading to reduced PPL score. Qualitative examples illustrating this are provided in Appendix M.3, while Ap-

pendix G further analyzes this effect by examining perplexity distributions. For NLI and S-BERT scores (Figure 15), the results align with the LLM judgment score and confirm our method's effectiveness.

## 5.3. Ablation Studies and Analysis

We conduct ablations on the logit bias $\beta$ and percentile $p$. We also evaluate our method on longer texts generated by a stronger LLM and validate the proxy suppression set $\widehat{\mathcal{G}}$ by analyzing detection bounds and comparing against random token selection. Appendix D compares computational cost across baselines and shows that our method remains efficient due to its simple operations. Unless otherwise specified, all experiments are conducted on the Llama-3.1-8B-Instruct model with the SIR watermarking scheme, following the experimental setup described in Section 5.1.

**Effect of logit bias $\beta$ and percentile $q$.** We vary $\beta$ from 0.0 to $-9.0$ with the percentile fixed at $q = 0.5$. Table 2 shows that without logit bias ($\beta = 0.0$, equivalent to vanilla paraphrasing), the ASR is low, but it increases as the absolute value of $\beta$ grows. This is consistent with Theorem 4.2: increasing the negative bias further suppresses green token sampling and thus lowers the overall probability of detection. However, larger negative values of $\beta$ gradually degrade text quality and require more iterations, as excessive bias restricts the token distribution too strongly.

Next, we vary $q$ from 0.0 to 0.9 with $\beta = -4.0$. Table 2 shows that when $q = 0$ (bias applied to all tokens in the watermarked text), ASR is moderately high, but text quality degrades slightly, and the number of iterations increases because many tokens are suppressed. As $q$ grows, the proxy set contains fewer tokens, and fewer are suppressed, so ASR drops since the watermark signal is not effectively removed, while text quality improves as the token distribution is less constrained.

**Long-text evaluation with a stronger LLM.** Longer texts make watermark detection easier, while OPT-1.3B, despite being commonly used for its high-entropy outputs (Kuditipudi et al., 2023), often produces low-quality text. We therefore evaluate on 500 samples of 600-token SIR-watermarked text generated by Qwen2.5-32B. Table 3 shows that BIRA remains effective in this stronger setting, whereas SIRA suffers severe quality degradation as masking becomes more disruptive on longer texts.

*Table 3.* Long-text evaluation on 600-token SIR-watermarked texts from Qwen2.5-32B. Additional results for other baselines and metrics are provided in Appendix E.3.

| Attack | TPR@FPR=1%↓ | LLM Judge↑ | PPL↓ |
|---|---|---|---|
| SIRA | 20.8% | 1.33 | 7.19 |
| BIRA (Ours) | 4.0% | 4.14 | 8.90 |

**Detection bound analysis.** To empirically evaluate the effectiveness of the proxy suppression set $\widehat{\mathcal{G}}$, we compute, for each of 500 samples under the Unigram watermark, the per-sample upper bound on detection probability implied by Theorem 4.2. For ease of analysis, we generate watermarked text with Llama-3.2-3B and apply the BIRA attack with Llama-3.1-8B, since the two models share the same tokenizer. For each sample, we calculate the average conditional green probability using the true green set $\mathcal{G}(\mathcal{W}_k)$

$$\bar{p} = \frac{1}{N} \sum_{n=1}^{N} \mathbb{E}\left[\mathbf{1}\{\tilde{y}^{(n)} \in \mathcal{G}(\mathcal{W}_k)\} \,\Big|\, \tilde{y}^{0:n-1}\right],$$

set $\hat{\delta} = \max\{0, p_\tau - \bar{p}\}$, and evaluate the bound $\exp\big(-\frac{1}{2}N\hat{\delta}^2\big)$. Figure 5 shows that BIRA produces substantially lower per-sample detection bounds than Vanilla for most samples. At the 90th percentile over 500 samples, BIRA's upper bound is $7.50 \times 10^{-2}$, compared to $7.97 \times 10^{-1}$ for Vanilla, indicating that the proxy suppression set $\widehat{\mathcal{G}}$ effectively reduces green-token sampling.

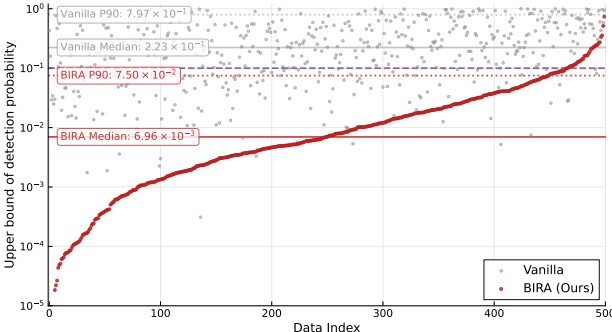

*Figure 5.* Detection upper bounds per sample from Theorem 4.2, sorted by BIRA. BIRA substantially reduces the upper bound on detection probability for most samples compared to Vanilla.

**Effectiveness of statistical signal suppression.** Table 4 presents the $z$-scores and detection thresholds $\tau$ for attacks against the SIR and Unigram schemes. Our method achieves the lowest $z$-scores among all baselines, effectively suppressing the watermark signal and rendering the text statistically indistinguishable from human writing. Additional results for other watermarking schemes are presented in Table 16, where our method consistently outperforms all baselines.

*Table 4.* $z$-score comparison of attacks on SIR and Unigram watermarking scheme.

| Watermark | SIR ($\tau = 0.2$) | Unigram ($\tau = 4.0$) |
|---|---|---|
| Vanilla | $0.19 \pm 0.10$ | $3.04 \pm 1.52$ |
| DIPPER-1 | $0.18 \pm 0.11$ | $3.63 \pm 1.65$ |
| DIPPER-2 | $0.14 \pm 0.11$ | $3.10 \pm 1.63$ |
| SIRA | $0.14 \pm 0.12$ | $1.63 \pm 1.38$ |
| BIRA (Ours) | $-0.06 \pm 0.09$ | $-0.34 \pm 1.61$ |

**Impact of token selection.** To evaluate the effectiveness of self-information–guided token selection for constructing the proxy suppression set $\widehat{\mathcal{G}}$, we vary the selection ratio from 0.1 to 0.9. At each ratio, we select the tokens with the highest self-information and compare against a proxy set of the same size formed by random token selection. As shown in Figure 6, applying a negative logit bias to self-information–guided tokens consistently yields stronger watermark evasion than random selection, indicating that self-information provides a practical signal for identifying and suppressing watermark traces.

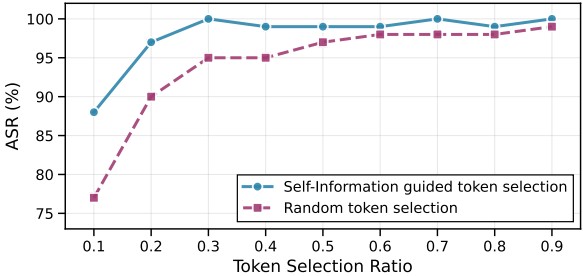

*Figure 6.* Comparison of ASR for self-information–guided token selection and random token selection.

## 6. Conclusion

This paper formalizes a theoretical analysis of rewriting attacks and, based on this insight, introduces the Bias-Inversion Rewriting Attack (BIRA), which applies a negative bias to a proxy suppression set during rewriting. We empirically demonstrate that BIRA is a practical, query-free attack that effectively erases watermark signals while maintaining high semantic fidelity. Our findings reveal critical limitations in current methods, underscoring the need for rigorous stress-testing and defenses robust to sophisticated rewriting.

## Acknowledgments

We are grateful to Professor He He for her valuable feedback and insightful discussions. This work was supported by the Institute of Information & Communications Technology Planning & Evaluation (IITP) grant funded by the Korea government(MSIT) (No.RS-2019-II191906, Artificial Intelligence Graduate School Program (POSTECH)); the IITP grant funded by the Korean government (MSIT) (No. RS-2024-00509258, Global AI Frontier Lab).

## Impact Statement

This paper aims to rigorously stress-test the robustness of current LLM watermarking schemes. While numerous studies demonstrate robustness against simple post-editing oper-

ations, our work reveals that these defenses are not yet sufficiently stress-tested against sophisticated black-box evasion. We demonstrate that a simple, practical rewriting attack can effectively eliminate watermark signals, highlighting a critical gap in current evaluation standards. Additionally, we acknowledge the potential risks associated with releasing a successful evasion method, as it could be used to bypass detection of misused content. However, we believe that transparently identifying vulnerabilities is a prerequisite for scientific progress. By exposing these limitations, our work serves as a necessary form of adversarial "red-teaming" aiming to foster the development of more robust defenses and contribute to the realization of truly responsible AI systems.

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

## A. Use of LLMs

In this paper, we only use LLMs to assist with text refinement such as trimming text, detecting grammatical errors, and correcting them.

## B. Proof of Theorem

*Proof of Lemma 4.1.* Fix $N \in \mathbb{N}$. By assumption there exists a nondecreasing $h : [0,1] \to \mathbb{R}$ with

$$Z(y; \mathcal{W}_k) = h\big(\hat{p}(y; \mathcal{W}_k)\big), \qquad \hat{p}(y; \mathcal{W}_k) = \frac{1}{N} \sum_{n=0}^{N-1} \mathbf{1}\{y_n \in \mathcal{G}(\mathcal{W}_k)\}.$$

The range of $\hat{p}$ is the grid $\mathcal{P}_N := \{0, 1/N, \dots, 1\}$. Define

$$p_\tau \;=\; \min\{\, p \in \mathcal{P}_N : h(p) \geq \tau \,\},$$

taking $p_\tau = 1$ if the set is empty and $p_\tau = 0$ if $h(p) \geq \tau$ for all $p \in \mathcal{P}_N$. Since $h$ is nondecreasing, for any $p, p' \in \mathcal{P}_N$ with $p \geq p_\tau > p'$ we have $h(p) \geq h(p_\tau) \geq \tau$ while $h(p') < \tau$. Therefore, for any $y$,

$$\mathcal{D}(y, \mathcal{W}_k) = \mathbf{1}\{Z(y; \mathcal{W}_k) \geq \tau\} = \mathbf{1}\{h(\hat{p}(y; \mathcal{W}_k)) \geq \tau\} = \mathbf{1}\{\hat{p}(y; \mathcal{W}_k) \geq p_\tau\}.$$

□

*Proof of Theorem 4.2.* By Lemma 4.1, for fixed $N$ there exists $p_\tau$ with $\mathcal{D}(y, \mathcal{W}_k) = \mathbf{1}\{\hat{p}(y; \mathcal{W}_k) \geq p_\tau\}$. Define the indicator variables and their conditional expectations:

$$X_n := \mathbf{1}\{\tilde{y}^{(n)} \in \mathcal{G}(\mathcal{W}_k)\}, \qquad p_n := \mathbb{E}[X_n \mid \mathcal{F}_{n-1}],$$

where $\mathcal{F}_n := \sigma(\tilde{y}^{0:n-1})$ is the natural filtration. The premise of the theorem is that the average conditional probability $\bar{p}_N := \frac{1}{N} \sum_{n=1}^{N} p_n$ satisfies $\bar{p}_N \leq p_\tau - \delta$.

Define the martingale difference sequence

$$D_n := X_n - p_n,$$

and the martingale

$$M_N := \sum_{n=1}^{N} D_n.$$

This is a martingale difference sequence since $\mathbb{E}[D_n \mid \mathcal{F}_{n-1}] = \mathbb{E}[X_n \mid \mathcal{F}_{n-1}] - p_n = p_n - p_n = 0$. Moreover, since $X_n \in \{0,1\}$ and $p_n \in [0,1]$, the increments are bounded in the interval $D_n \in [-1,1]$.

We can relate the empirical green rate $\hat{p}(\tilde{y}; \mathcal{W}_k)$ to the martingale $M_N$:

$$\hat{p}(\tilde{y}; \mathcal{W}_k) = \frac{1}{N} \sum_{n=1}^{N} X_n = \frac{1}{N} \sum_{n=1}^{N} (D_n + p_n) = \frac{M_N}{N} + \bar{p}_N.$$

Thus, the detection event occurs iff:

$$\hat{p}(\tilde{y}; \mathcal{W}_k) \geq p_\tau \iff \frac{M_N}{N} + \bar{p}_N \geq p_\tau \iff M_N \geq N(p_\tau - \bar{p}_N).$$

Using the premise that $\bar{p}_N \leq p_\tau - \delta$, we have $p_\tau - \bar{p}_N \geq \delta$. Therefore,

$$\Pr(\hat{p}(\tilde{y}; \mathcal{W}_k) \geq p_\tau) \leq \Pr(M_N \geq N\delta).$$

By the Azuma–Hoeffding inequality, for increments $D_n$ bounded in an interval of range $1 - (-1) = 2$,

$$\Pr(M_N \geq N\delta) \leq \exp\left( -\frac{2(N\delta)^2}{\sum_{n=1}^{N} 2^2} \right) = \exp\left( -\frac{2N^2 \delta^2}{4N} \right) = \exp\left( -\frac{N\delta^2}{2} \right),$$

which yields the claim.

□

## C. Details of the Text Degeneration Detection Function

---

**Algorithm 2** Text Degeneration Detection

---

1: **Input:** Paraphrased text $\tilde{y} = [\tilde{y}^{(0)}, \ldots, \tilde{y}^{(L-1)}]$; collapse window $h \in \mathbb{N}$; collapse threshold $\rho \in (0, 1]$.
2: $m \leftarrow \min(h, L)$
3: $W \leftarrow [\tilde{y}^{(L-m)}, \ldots, \tilde{y}^{(L-1)}]$          *(last m tokens)*
4: $U \leftarrow \{\text{id}(u) \mid u \in W\}$          *(distinct token ids)*
5: **if** $|W| = 0$ **then**
6:      **return** `False`
7: **end if**
8: $r \leftarrow |U|/|W|$          *(distinct 1-gram ratio)*
9: **if** $r < \rho$ **then**
10:      **return** `True`          *(degeneration detected)*
11: **else**
12:      **return** `False`
13: **end if**

---

For paraphrasing, we set the maximum generation length of the LLM to 1,500 tokens. We observed that rewritten text is typically generated normally when no degeneration occurs. However, when degeneration does occur, the text begins normally but then suddenly repeats the same phrase until the maximum token limit is reached, as shown in Appendix M.2. In the degenerated samples we examined, both the starting point of the repetition and the length of the repeated phrase varied. To ensure a sufficient detection window, we chose a large maximum generation length and a window size of $h = 450$ tokens. We set the threshold $\rho = 0.25$, meaning that if more than 75% of the tokens within the detection window are duplicates (which is not normal for natural text), the text is considered largely repetitive and redundant.

## D. Computational Efficiency Evaluation

*Table 5.* Average execution time (in seconds) for different attacks.

| Attack | Time |
|---|---|
| Vanilla | $5.52 \pm 2.82$ |
| DIPPER | $9.73 \pm 2.24$ |
| SIRA | $8.57 \pm 2.03$ |
| BIRA (Ours) | $7.95 \pm 9.01$ |

To assess computational overhead, we measured the average execution time per attack over 500 samples using the KGW watermark under the setup in Section 5.1. All experiments were conducted on a single A6000 GPU, except for DIPPER built on T5-XXL (Raffel et al., 2020), which required two GPUs. As shown in Table 5, Vanilla is the most efficient baseline since it introduces no additional overhead. BIRA is the next most efficient, though it exhibits higher variance. This is caused by its adaptive bias procedure, which is designed to prevent text degeneration. This procedure was triggered in only 2.6% of samples, and those rare cases had a much longer average runtime of 66.81 seconds because repeated generation continued until the maximum length is reached. By contrast, the vast majority of samples (97.4%) completed in a single iteration with an average of 6.38 seconds, accounting for BIRA's overall efficiency despite the variance introduced by a few outliers.

## E. Extended Experimental Results

### E.1. Additional Dataset Results

To strengthen the evaluation of our method, we conduct additional experiments on two datasets, Dolly CW (Conover et al., 2023) and DBPEDIA Class (Raffel et al., 2020), using 100 examples and the experimental setup described in Section 5.1 with the Llama 3.1-8B model. We evaluate attack performance using the default hyperparameters, report TPR at FPR levels of 1% and 10% to mitigate the effect of a fixed threshold, and include text quality evaluation.

Table 6 and Table 7 show that our method consistently outperforms the baselines under the default settings. Figure 9 and

*Table 6.* Comparison of watermarking robustness under different attack methods on the Dolly CW dataset.

| Attack \ Watermark | KGW | Unigram | UPV | EWD | DIP | SIR | EXP |
|---|---|---|---|---|---|---|---|
| Vanilla (Llama-3.1-8B) | 81.0% | 65.0% | 70.0% | 78.0% | 99.0% | 49.0% | 68.0% |
| DIPPER-1 | 94.0% | 61.0% | 73.0% | 96.0% | 99.0% | 62.0% | 87.0% |
| DIPPER-2 | 98.0% | 74.0% | 83.0% | 94.0% | 100.0% | 66.0% | 95.0% |
| SIRA (Llama-3.1-8B) | 97.0% | 91.0% | 90.0% | 99.0% | 99.0% | 73.0% | 94.0% |
| **BIRA (Llama-3.1-8B, ours)** | **100.0%** | **99.0%** | **99.0%** | **100.0%** | **99.0%** | **98.0%** | **97.0%** |

*Table 7.* Comparison of watermarking robustness under different attack methods on the DBPEDIA Class dataset.

| Attack \ Watermark | KGW | Unigram | UPV | EWD | DIP | SIR | EXP |
|---|---|---|---|---|---|---|---|
| Vanilla (Llama-3.1-8B) | 76.0% | 63.0% | 61.0% | 83.0% | 97.0% | 46.0% | 64.0% |
| DIPPER-1 | 87.0% | 52.0% | 75.0% | 92.0% | 100.0% | 67.0% | 83.0% |
| DIPPER-2 | 93.0% | 63.0% | 76.0% | 95.0% | 100.0% | 80.0% | 96.0% |
| SIRA (Llama-3.1-8B) | 95.0% | 90.0% | 90.0% | 99.0% | 97.0% | 71.0% | 88.0% |
| **BIRA (Llama-3.1-8B, ours)** | **98.0%** | **100.0%** | **96.0%** | **99.0%** | **100.0%** | **99.0%** | **95.0%** |

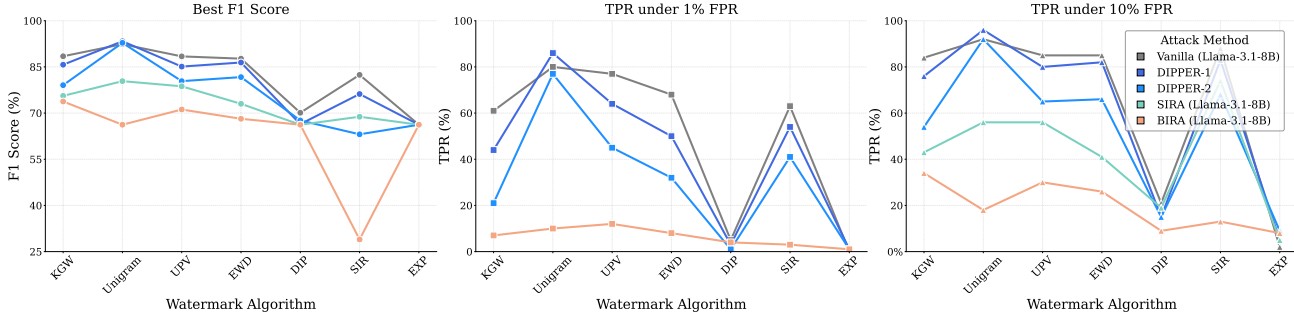

*Figure 7.* Comparison of detection performance with adjusted thresholds across watermarking algorithms on the Dolly CW dataset. We show the best F1 score (↓) and TPR (↓) at FPR of 1% and 10%.

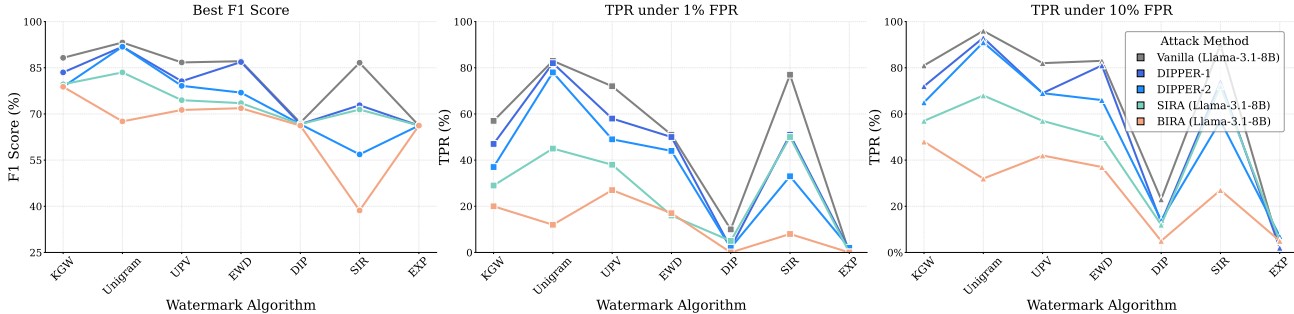

*Figure 8.* Comparison of detection performance with adjusted thresholds across watermarking algorithms on the DBPEDIA dataset. We show the best F1 score (↓) and TPR (↓) at FPR of 1% and 10%.

Figure 10 further demonstrate that our method achieves superior performance under the adaptive threshold setup.

For text quality evaluation, our method preserves semantic fidelity more effectively than the strong baselines DIPPER and SIRA across the five metrics, consistent with the trends observed in the main results (Figure 3).

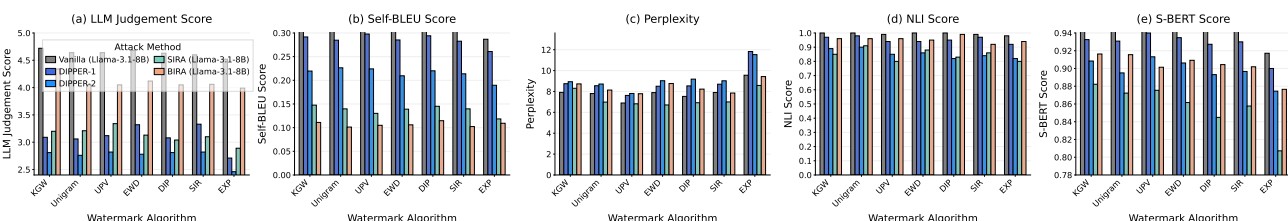

*Figure 9.* Comparison of text quality across different attacks for various watermarking methods, evaluated using LLM judgment score (↑), Self-BLEU (↓), and Perplexity (↓) on the Dolly CW dataset.

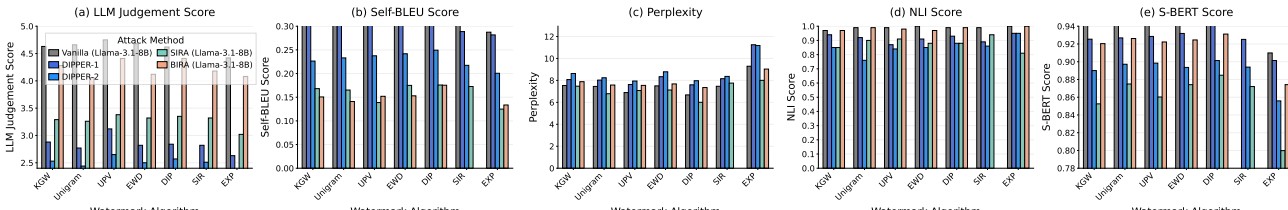

*Figure 10.* Comparison of text quality across different attacks for various watermarking methods, evaluated using LLM judgment score (↑), Self-BLEU (↓), and Perplexity (↓) on the DBPEDIA dataset.

## E.2. Additional Robustness Evaluations

### E.2.1. APPLICABILITY TO SENTENCE-LEVEL WATERMARKING

Our method is primarily designed for token-level watermarking schemes, since they are fundamental and dominant paradigms. To strengthen robustness against paraphrasing, recent works (Hou et al., 2024; Dabiriaghdam & Wang, 2025) have proposed more robust sentence-level watermarking schemes, albeit with much higher computational cost. However, these schemes are conceptually similar to token-level watermarking, so the core principle behind BIRA naturally extends to these sentence-level methods as well.

Conceptually, sentence-level watermarking schemes (Hou et al., 2024; Dabiriaghdam & Wang, 2025) replace the "green/red" token distinction with a "valid/invalid" region in sentence embedding space. Let $s_n$ denote the $n$-th sentence and $e_n = f_{\text{em}}(s_n)$ its embedding under an embedding model $f_{\text{em}}$. During generation, these methods repeatedly regenerate $s_{n+1}$ until the similarity

$$\text{sim}(e_n, e_{n+1})$$

falls within a predefined interval $[a, b]$, and detection is based on counting how many sentence pairs satisfy this validity condition. This is directly analogous to token-level watermarks that count how many tokens fall into a green token set.

**Extension of Theorem 4.2 to Sentence-Level Watermarking.** Our Theorem 4.2 can be extended by redefining the random variable from an indicator of sampling a green token to an indicator of a valid sentence pair. Specifically, instead of

$$Z_n = \mathbf{1}\{\tilde{y}^{(n)} \in \mathcal{G}(\mathcal{W}_k)\},$$

we consider

$$Z_n = \mathbf{1}\big[\text{sim}(f_{\text{em}}(s_n), f_{\text{em}}(s_{n+1})) \in [a, b]\big],$$

and the detector thresholds the empirical average

$$\hat{p}_{\text{sent}} = \frac{1}{N} \sum_{n=1}^{N} Z_n.$$

For the rewriting attack, let $\tilde{s}_n$ denote the rewritten sentences and let $\tilde{f}_{\text{em}}$ denote an auxiliary embedding. We then define

$$\tilde{Z}_n = \mathbf{1}\big[\text{sim}(\tilde{f}_{\text{em}}(\tilde{s}_n), \tilde{f}_{\text{em}}(\tilde{s}_{n+1})) \in [\tilde{a}, \tilde{b}]\big].$$

During rewriting, we repeatedly generate paraphrases $\tilde{s}_{n+1}$ until they satisfy

$$\text{sim}\big(\tilde{f}_{\text{em}}(\tilde{s}_n), \tilde{f}_{\text{em}}(\tilde{s}_{n+1})\big) \notin [\tilde{a}, \tilde{b}],$$

*Table 8.* Comparison of ASR and text quality across different attacks against SimMark sentence-level watermarking.

| Method | ASR (↑) | LLM Judgement (↑) | Self-BLEU (↓) | PPL (↓) | NLI (↑) | S-BERT (↑) |
|---|---|---|---|---|---|---|
| Vanilla (Llama-3.1-8B) | 0.544 | 4.752 | 0.228 | 8.191 | 0.958 | 0.904 |
| DIPPER-1 | 0.054 | 3.454 | 0.332 | 8.876 | 0.930 | 0.908 |
| DIPPER-2 | 0.082 | 3.036 | 0.238 | 9.505 | 0.856 | 0.878 |
| SIRA (Llama-3.1-8B) | 0.872 | 3.450 | 0.102 | 7.988 | 0.728 | 0.830 |
| BIRA (Llama-3.1-8B, ours) | 0.826 | 4.110 | 0.060 | 9.665 | 0.862 | 0.857 |

*Table 9.* Robustness comparison for long-text evaluation on 600-token SIR-watermarked texts generated by Qwen2.5-32B.

| Method | TPR@FPR=1%↓ | TPR@FPR=10%↓ | Best F1↓ | LLM Judge↑ | PPL↓ | Self-BLEU↓ |
|---|---|---|---|---|---|---|
| Vanilla (Llama-3.1-8B) | 57.7% | 81.8% | 86.8% | 4.65 | 7.13 | 0.24 |
| DIPPER-1 | 41.2% | 78.6% | 86.5% | 3.70 | 8.50 | 0.25 |
| DIPPER-2 | 25.2% | 60.2% | 82.3% | 3.25 | 8.90 | 0.19 |
| SIRA (Llama-3.1-8B) | 20.8% | 48.4% | 73.8% | 1.33 | 7.19 | 0.02 |
| BIRA (Llama-3.1-8B, Ours) | 4.0% | 11.0% | 29.4% | 4.14 | 8.90 | 0.05 |

which effectively reduces the probability that $\tilde{Z}_n = 1$. Following the same logic as in Theorem 4.2, ensuring

$$\frac{1}{N} \sum_{n=1}^{N} \mathbb{E}[\tilde{Z}_n] \leq p_\tau - \delta$$

for some $\delta > 0$ yields an analogous exponential upper bound on the detector's success probability.

Thus, although our current theorem and implementation operate at the token level, the theoretical framework and BIRA's principle extend naturally to sentence-level watermarking. An interesting direction for future work is to adapt our method to sentence-level watermarking in practice, where a key question is how well the adversary's embedding model $\tilde{f}_{em}$ aligns with the embedding model $f_{em}$ used in generating watermarked text, and how to estimate the unknown validity interval $[\tilde{a}, \tilde{b}]$.

**Experimental results of BIRA on sentence-level watermarking.** To evaluate the effectiveness of our token-level BIRA attack against sentence-level watermarking, we conduct experiments under the setup in Section 5.1 using the Llama-3.1-8B model with the SimMark watermarking scheme (Dabiriaghdam & Wang, 2025), which is a state-of-the-art sentence-level watermark. For detection, we use a $z$-score threshold of $5.03$, which yields an FPR of $1\%$, and adopt the hyperparameter setting recommended by Dabiriaghdam & Wang (2025), using the interval $[0.68, 0.76]$ for cosine similarity on the C4 dataset.

Table 8 shows that BIRA still achieves strong attack performance. Sentence-level watermarking relies on consistent sentence segmentation between generation and detection, whereas BIRA performs document-level paraphrasing that can merge or split sentences, thereby weakening this assumption. In contrast, DIPPER-1 and DIPPER-2 perform poorly against SimMark because they paraphrase each sentence independently, preserving the sentence segmentation exploited by sentence-level watermarks. These results suggest that current robustness evaluations for sentence-level watermarks are insufficiently rigorous and can create a misleading impression of robustness. When further evaluated on another sentence-level watermarking scheme, PMARK (Huo et al., 2025), our method achieves a TPR of 0.04 at FPR= $1\%$ while maintaining an LLM judge score of 4.35, demonstrating its effectiveness.

### E.3. Evaluation on Longer Texts with a Stronger LLM

Table 9 shows that our method remains effective on 600-token SIR-watermarked texts, outperforming baselines while preserving semantic fidelity. In contrast, SIRA degrades substantially as text length increases: its masking process becomes increasingly disruptive, hindering LLM infilling and leading to poor semantic preservation.

*Table 10.* Watermark performance of low-entropy SIR, which embeds watermark signals only below the 30th/50th-percentile entropy thresholds estimated from human-written texts. Achieving high detectability requires a much larger logit bias $\gamma$, substantially degrading text quality as reflected by higher PPL.

| Setting | TPR@FPR=1% (↑) | TPR@FPR=10% (↑) | PPL (↓) |
|---|---|---|---|
| Standard-SIR ($\gamma = 1.0$) | 0.998 | 1.000 | 11.95 |
| Low-SIR ($\gamma = 1.0$) | 0.098 / 0.596 | 0.368 / 0.884 | 9.87 / 10.64 |
| Low-SIR ($\gamma = 2.0$) | 0.612 / 0.988 | 0.896 / 1.000 | 12.07 / 14.08 |
| Low-SIR ($\gamma = 4.0$) | 0.980 / 1.000 | 1.000 / 1.000 | 16.72 / 19.13 |

*Table 11.* BIRA performance against Low-SIR using the 50th-percentile entropy threshold. This variant yields slightly higher TPR, but the absolute TPR remains low, and the text-quality cost is substantial.

| Setting | TPR@FPR=1% (↓) | TPR@FPR=10% (↓) | PPL (↓) | LLM Judge (↑) |
|---|---|---|---|---|
| Standard-SIR | 0.012 | 0.114 | 10.19 | 4.24 |
| Low-SIR ($\gamma = 1.0$) | 0.096 | 0.230 | 10.50 | 4.26 |
| Low-SIR ($\gamma = 2.0$) | 0.134 | 0.310 | 10.93 | 4.05 |
| Low-SIR ($\gamma = 4.0$) | 0.206 | 0.382 | 11.44 | 3.86 |

### E.4. Evaluation on Localized Detector

Unlike standard detectors that score the entire text, WinMax (Kirchenbauer et al., 2024b) detects localized watermarked regions by searching for the contiguous token span with the highest $z$-score. To evaluate BIRA under this detector, we use 500 KGW-watermarked 600-token texts generated by Qwen2.5-32B and rewritten by Llama-3.1-8B. BIRA achieves a TPR of 5.0% at 1.0% FPR, showing that its global proxy suppression set effectively suppresses watermark signals across the entire text.

### E.5. Analysis of Failure Modes

We examine two potential defenses against high-surprisal token suppression: (1) shifting watermark signals toward lower-entropy tokens and (2) distributing signals more evenly across token positions using a larger logit bias.

**Shifting signals to lower-entropy tokens.** A watermark can, in principle, place more signal on lower-entropy tokens. However, these tokens offer fewer plausible alternatives, so reliable detection requires stronger biases that substantially degrade text quality. We implement a low-entropy variant by applying the watermark bias only to low-entropy positions. Following the setup in Section 5.1, we compute token entropy on 500 natural-text samples, estimate the 30th- and 50th-percentile entropy thresholds for each sentence, average them across samples, and apply detection only to tokens below the selected threshold.

Table 10 reports watermark-only performance under the 30th/50th-percentile thresholds, with standard SIR as a reference. Compared with standard SIR, low-entropy watermarking requires a much stronger bias to achieve reliable detection, since low-entropy positions allow less perturbation and provide less total signal to the detector. As the bias increases, PPL rises sharply, and text quality degrades, often producing grammatical errors, unusual spelling, and repetitive phrases. Thus, shifting the signal to low-entropy positions leads to a worse detectability–quality tradeoff rather than a practical defense.

We then evaluate BIRA against the 50th-percentile low-entropy variant, which offers the best tradeoff. As shown in Table 11, this variant yields slightly higher post-rewriting TPR than standard SIR, but the absolute TPR remains low. Although BIRA less directly targets lower-entropy signals, they are sparse and still weakened by paraphrasing, which changes both the surrounding context and the token subset selected at detection time. Rewriting also removes many artifacts introduced by large biases, further reducing the detector signal. Overall, the robustness gain is modest, while the text-quality cost is substantial.

**Distributing signals more evenly across positions.** We next consider increasing the SIR watermark bias $\gamma$, which strengthens watermark evidence across more token positions, including lower-entropy ones. Table 12 shows that detectability remains high as $\gamma$ increases, but PPL rises substantially, indicating greater distortion of the model distribution.

*Table 12.* Watermark performance of SIR with larger watermark bias $\gamma$. As $\gamma$ increases, text quality degrades substantially, as reflected by higher PPL.

| Setting | TPR@FPR=1% ($\uparrow$) | TPR@FPR=10% ($\uparrow$) | PPL ($\downarrow$) |
|---|---|---|---|
| Standard-SIR ($\gamma = 1.0$) | 0.998 | 1.000 | 11.95 |
| Standard-SIR ($\gamma = 2.0$) | 1.000 | 1.000 | 15.70 |
| Standard-SIR ($\gamma = 4.0$) | 1.000 | 1.000 | 19.16 |

*Table 13.* BIRA performance against SIR with larger watermark bias $\gamma$. Larger $\gamma$ slightly increases TPR, but the absolute TPR remains low and comes with substantial text-quality degradation.

| Setting | TPR@FPR=1% ($\downarrow$) | TPR@FPR=10% ($\downarrow$) | PPL ($\downarrow$) | LLM Judge ($\uparrow$) |
|---|---|---|---|---|
| Standard-SIR ($\gamma = 1.0$) | 0.012 | 0.114 | 10.19 | 4.24 |
| Standard-SIR ($\gamma = 2.0$) | 0.032 | 0.138 | 10.42 | 3.96 |
| Standard-SIR ($\gamma = 4.0$) | 0.064 | 0.210 | 10.71 | 3.72 |

Table 13 reports BIRA results on these variants. Larger $\gamma$ makes BIRA slightly less effective, but the improvement in robustness is limited and comes at a clear quality cost. The lower LLM-judge scores reflect poorer source text quality, which makes faithful rewriting harder. Thus, spreading the signal more broadly is not a practical defense; it mainly trades text quality for modest robustness gains.

Overall, both failure modes reduce BIRA's effectiveness only modestly, while requiring stronger watermark biases and noticeably degrading text quality.

## F. Discussion and Future Directions

Although BIRA is highly effective and preserves semantics better than existing baselines, further improving semantic preservation remains an important direction. One promising approach is to replace the global proxy suppression set and uniform negative logit bias with context-adaptive or position-wise biasing, which may reduce distributional distortion while maintaining strong watermark suppression.

Another interesting direction is to study mixed-domain settings, where natural language and code snippets appear in the same text. In such cases, a token may be high-entropy in one context but low-entropy in another, making context-aware suppression especially important.

---

**Human written text in C4 dataset with PPL 25.4**

Yes, we all have separate inboxes. There is no giant inbox where all messages to moderators go. So if we either don't understand a PM we get from a mod or admin we're allowed to ask in PM? Am I allowed to say I have a really hard time reading really long PMs? What I describe as motorway length PMs cause I get halfway then my brain turns it to word salad and goes into information overload? Maybe you need to clarify the rule so not wrong things can be read into it. I wish I had been that lucky. I always get anxiety when I get pm from a monitor. Ok! I didn't know that! Exactly I just discover this myself! Don't let Turtleboy fool you! He's bad to the bone! I know many on the boards here are very sensitive & struggle with anxiety, but everyone has a completely anonymous screen name. No one knows who "emgreen"

---

*Figure 11.* An example of human written text in the C4 dataset with high PPL

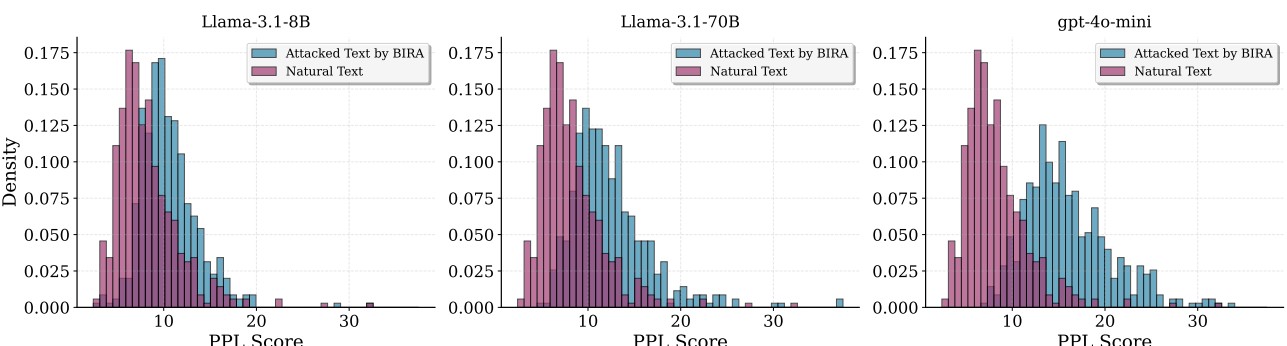

*Figure 12.* Overlap between the PPL distributions of natural text and BIRA-attacked text for KGW watermarking.

## G. Analysis of Perplexity Increase under BIRA

In this section, we analyze two concerns raised by the increased PPL under the BIRA attack. While our method slightly increases the PPL of attacked text for Llama-3.1-8B and Llama-3.1-70B, the PPL for GPT-4o-mini increases more noticeably. This raises two questions: (1) does this increase lead to a perceptible degradation of readability in real-world scenarios, and (2) can PPL be used as a post-hoc signal to flag attacked text?

**Human evaluation of attacked text with high PPL.** Qualitative analysis in Appendix M.3 shows that attacked texts with high PPL remain grammatical and reasonably readable despite their elevated perplexity scores. Our evaluation suggests that this PPL increase may stem from stylistic factors such as overuse of hyphens or particular synonym choices. Crucially, we also observe that human-written text from the C4 dataset can exhibit similarly high PPL, particularly in informal or conversational sentences (Figure 11). This supports the view that PPL is not a perfect proxy for human-perceived quality in real-world scenarios.

**Can PPL be used as a post-hoc signal to flag attacked text?** Perplexity alone is not a sufficiently reliable detection signal, due to its significant overlap between natural and attacked text. As illustrated in Figure 12, the PPL distributions of attacked text by BIRA for KGW watermarking and natural text in C4 dataset overlap substantially for Llama-3.1-8B and Llama-3.1-70B, making PPL ineffective for discriminating between them. For GPT-4o-mini, the BIRA distribution is more shifted toward higher PPL, but there is still considerable overlap with natural text. We suspect that part of this shift is caused by the mismatch between GPT-4o-mini's tokenizer and the auxiliary tokenizer (Llama-3.1-3B) used to evaluate token self-information, and GPT-4o-mini's writing property that generate lots of hyppen.

Additionally, unlike statistical watermarking, which provides formal guarantees via a calibrated test statistic, PPL-based detection has no such theoretical grounding. Thus, although designing BIRA variants that lower PPL while preserving attack strength is interesting future work, perplexity remains a heuristic and lacks the robustness needed for a reliable post-hoc detection mechanism.

*Table 14.* Comparison of watermarking robustness under different attack methods with ADA fine-tuned on KGW watermark.

| Attack \ Watermark | KGW | Unigram | UPV | EWD | DIP | SIR | EXP |
|---|---|---|---|---|---|---|---|
| Vanilla (Llama-3.1-8B) | 88.8% | 73.4% | 73.4% | 92.6% | 99.8% | 54.0% | 80.6% |
| Dipper-1 | 93.8% | 61.2% | 80.6% | 92.8% | 99.4% | 55.6% | 90.8% |
| Dipper-2 | 97.2% | 71.8% | 85.4% | 96.6% | 99.2% | 70.4% | 97.2% |
| ADA (Llama-3.1-8B) | 98.0% | 90.2% | 88.4% | 99.2% | 100.0% | 71.0% | 96.6% |
| SIRA (Llama-3.1-8B) | 98.8% | 95.0% | 87.6% | 99.8% | 99.6% | 72.8% | 95.2% |
| **BIRA (Llama-3.1-8B, ours)** | **99.8%** | **99.4%** | **99.8%** | **100.0%** | **100.0%** | **99.6%** | **99.8%** |

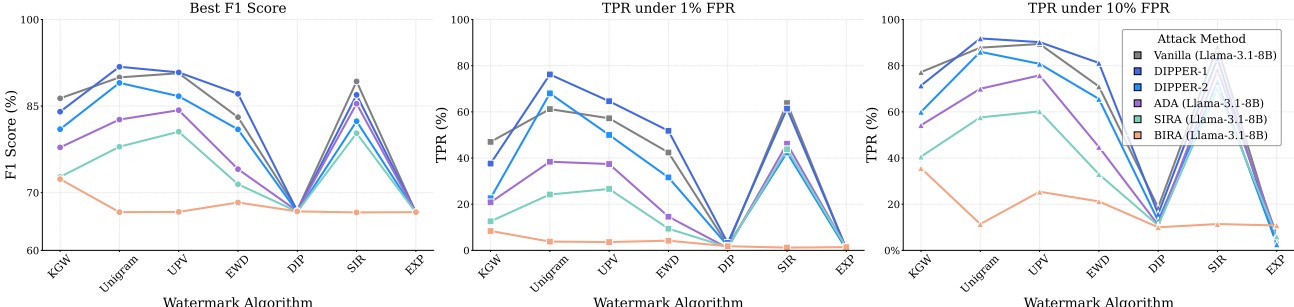

*Figure 13.* Comparison of detection performance across watermarking algorithms with ADA fine-tuned on the KGW watermark. We show the best F1 score (↓) and TPR (↓) at FPR of 1% and 10%.

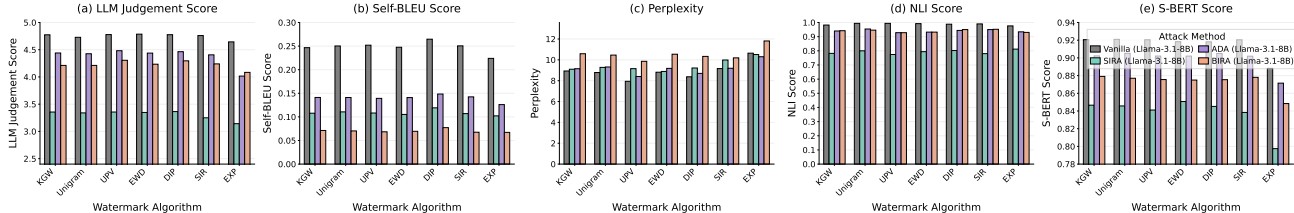

*Figure 14.* Comparison of text quality across different attacks with ADA fine-tuned on the KGW watermark, evaluated using LLM judgment score (↑), Self-BLEU (↓), and Perplexity (↓).

# H. Comparison with Adaptive Attacks

In this work, our main evaluation focuses on query-free attacks without knowledge of the watermarking scheme. To further assess the effectiveness of our method, we also compare against the adaptive attack (ADA) of Diaa et al. (2024), which operates in a less restrictive threat model: ADA assumes the watermarking scheme is known, fine-tunes a paraphrasing model specifically to that watermark, and then relies on transferability to unseen watermarking schemes. For this comparison, we follow the setup in Section 5.1 using Llama-3.1-8B fine-tuned on KGW watermarking for ADA.

Table 14 and Figure 13 show that SIRA and BIRA outperform ADA in attack success, even though they operate under a more restrictive threat model. For text quality, however, Figure 14 shows that ADA attains higher semantic fidelity in terms of LLM judgment score. These results indicate that BIRA is a substantially stronger attack, with a slight trade-off in semantic fidelity.

# I. Tokenizer Mismatch Analysis for GPT-4o-mini

*Table 15.* Ablation study of auxiliary models for estimating high-surprisal tokens under tokenizer mismatch with GPT-4o-mini.

| Metric | Average |
|---|---|
| TPR@FPR=1%($\downarrow$) | $0.019 \pm 0.005$ |
| LLM Judge ($\uparrow$) | $4.72 \pm 0.03$ |
| Iterations ($\downarrow$) | $1.02 \pm 0.01$ |

Since GPT-4o-mini does not expose logits, we use an auxiliary model to identify high-surprisal tokens and then re-tokenize them with the GPT tokenizer. To evaluate the effect of this tokenizer mismatch, we follow the setup in Section 5.1 and vary only the auxiliary model across Llama-3.2-3B, Llama-3.1-8B, Qwen3-8B, and Phi-4-mini. Table 15 shows that performance is stable across auxiliary models. For Llama-3.2-3B and Llama-3.1-8B, which share the same tokenizer, the Jaccard similarity between their selected token sets is 0.88, indicating substantial overlap.

# J. Prompt for Semantic Judgment (GPT)

> **LLM-as-a-Judge Prompt**
>
> You are an impartial evaluator.
> You will receive: 1) **ORIGINAL**: the source text 2) **PARAPHRASE**: a rewritten version of the original
> **Your task:** Judge how well the PARAPHRASE preserves the ORIGINAL's semantic meaning. Ignore style, tone, formality, phrasing, length, and order of information.
> **What to check (do not output your analysis):** - Core propositions and claims are preserved. - Entities, numbers, dates, units, polarity/negation, modality, and causal/temporal relations match. - No contradictions; no key facts dropped or altered. - Added details that do not change meaning should not be penalized.
> **Rating scale (1–5):** [5] Complete preservation — bidirectional entailment; no contradictions; all key facts preserved. [4] High preservation — meaning preserved with at most minor nuances/omissions; no factual changes. [3] Moderate — main idea preserved, but some important details/relations are missing or changed. [2] Low — only basic topic overlaps; several important changes/omissions or contradictions. [1] Minimal/none — meaning is fundamentally different or unrelated.
> **Special cases:** - If specificity differs, score based on whether the essential meaning is preserved. - Any mismatch in numbers, named entities, dates, or negation is a serious error.
> **Output Format:** - Provide *only* a number in square brackets. No explanations, additional text, or commentary. - Correct format: [3] - Incorrect format: "The score is [3]" | "I rate this [3]" | "[3] - good preservation"

**User Prompt:**

> Here's the ORIGINAL: [Watermarked text]
>
> Here's the PARAPHRASE: [Attack text]

## K. Rewriting Prompt

---

**Paraphrasing Prompt**

You are an expert paraphrasing system designed to rewrite text while maintaining complete semantic fidelity. Follow these precise guidelines:

**CORE REQUIREMENTS:** - Preserve 100% of the original meaning and information content - Maintain identical factual accuracy, tone, and intent - Keep the same level of formality/informality as the source - Retain all specific details, numbers, names, and technical terms exactly as provided

**TRANSFORMATION RULES:** - Use synonyms and alternative phrasings where semantically equivalent - Vary sentence structure and length for natural flow - Reorganize clauses and phrases without altering logical relationships - Maintain coherence and readability in the target language - Ensure each transformed sentence conveys the same scope and specificity as the original

**STRICT PROHIBITIONS:** - Do not add interpretations, assumptions, elaborations, or external knowledge - Do not introduce information not explicitly present in the original text - Do not amplify, exaggerate, or minimize any claims or statements - Do not omit any information present in the original - Do not change the perspective, viewpoint, or stance - Do not include explanations, commentary, or meta-text - Do not expand on implicit meanings or draw inferences - Do not add contextual information or background details

**CONTENT BOUNDARIES:** - Work only with information explicitly stated in the source text - If the original is vague or general, keep the paraphrase equally vague or general - If the original is specific, maintain that exact level of specificity - Do not fill in gaps or provide additional details, even if they seem logical

**OUTPUT FORMAT:** - Provide only the paraphrased text - Match the original format (paragraphs, lists, etc.) - No prefacing remarks, explanations, or additional content

**QUALITY CHECK:** Before outputting, verify that: 1) Someone reading only your paraphrase would understand exactly the same information as someone reading the original text 2) No new information has been introduced 3) No original information has been lost or altered 4) The scope and specificity remain identical

---

## L. $z$-score comparison of attacks on different watermarking schemes

*Table 16.* $z$-score comparison of attacks on different watermarking methods.

| Watermark | KGW-1 | Unigram | UPV | EWD | DIP | SIR |
|---|---|---|---|---|---|---|
| Vanilla | $2.40 \pm 1.32$ | $3.04 \pm 1.52$ | $3.24 \pm 1.23$ | $2.08 \pm 1.32$ | $0.16 \pm 0.52$ | $0.19 \pm 0.10$ |
| DIPPER-1 | $2.09 \pm 1.23$ | $3.63 \pm 1.65$ | $3.33 \pm 1.28$ | $2.37 \pm 1.18$ | $0.17 \pm 0.53$ | $0.18 \pm 0.11$ |
| DIPPER-2 | $1.66 \pm 1.19$ | $3.10 \pm 1.63$ | $2.97 \pm 1.65$ | $1.81 \pm 1.23$ | $0.06 \pm 0.63$ | $0.14 \pm 0.11$ |
| SIRA | $1.05 \pm 1.23$ | $1.63 \pm 1.38$ | $2.19 \pm 1.21$ | $0.83 \pm 1.13$ | $0.02 \pm 0.52$ | $0.14 \pm 0.12$ |
| BIRA | $0.93 \pm 1.11$ | $-0.34 \pm 1.61$ | $0.94 \pm 1.22$ | $0.43 \pm 1.07$ | $-0.03 \pm 0.51$ | $-0.06 \pm 0.09$ |

# M. Qualitative examples

## M.1. Examples of watermarked texts and attacked texts

**Watermarked Text By KGW ($z$-score: 10.33)**

Was it a surprise to you that you were given the arts and culture position? No, there is no surprise when you are a cadre. And it was a great honor. I enjoy serving my country in that capacity. It's what I really love. If anything, it was good to hear you say no once more; given that it was the first official rejection in your official career. Did anything take priority for you that the position didn't? Yes. One of those in-laws I wanted to stay with a little too much priority. But I'm glad to know you asked. I certainly wasn't giving up on helping my country in any capacity, that's not me at all. I just didn't want to go at it one more time. Once you go you will know how to say no. I had already been told in 2008 to hold off on being an officer until I got my undergraduate degree, that was one such instance. There's your answer. Ha, that is great. And good luck, I'm sure you'll do something great.

**Attacked Text by BIRA ($z$-score: 2.60)**

Was receiving the arts and culture position unexpected for you? Not at all, since I am part of the cadres. It was actually a huge honor. I truly love serving my country in this role. Given that this was your first formal rejection, I was glad to see you decline. Was there something specific that took precedence over this opportunity? One thing did - my desire to spend time with my in-laws took higher priority. I appreciate you asking. Please know I am still committed to contributing to my country's efforts. I simply chose not to pursue this particular path. Saying "no" becomes easier with time. For example, I was advised back in 2008 to delay becoming an officer until I finished my undergrad studies. That's excellent. I wish you all the best and am confident you'll achieve great things.

**Watermarked Text by SIR ($z$-score: 0.57)**

MIDDLETOWN , NJ - The Middletown Township Public Library and the Township of Middletown are pleased to announce the 2014 Fall Book & Family Fun Series Fall Book Club Pick: "Where The Crawdads Sing" by Delia Owens Sometime in late November or early December , library volunteers will be coming to your neighborhood with book bags and taking you to see and talk with some of our favorite children's authors , including Ann M Robbins, Diane Trela , Mary Pope Osborne , Beverly Cleary, Eric Carle , Chris Van Allsburg and more ! This fall , your friendly neighborhood library will offer two fun , free events ! Saturday , November 29 from 10 to 11 am join us for cookies , coffee and a reading of "Where The Kids Play ." Meet author and illustrator Ann M Robbins and hear her read from her new book , where her latest creations include Miss Mabel and two other kitty mascots named Fat and Happy . For those who would prefer reading to eating , we will offer a reading from our previous fall book club favorite , "Where The Bats Don ' t

**Attacked Text by BIRA ($z$-score: 0.11)**

MIDDLETOWN , NJ - Middletown Township Public Library along with the Township of Middletown is excited to announce the selection for the 2014 Fall Book & Family Fun Series : "Where the Crawdads Sing " by Delia Owens . In either November or early December , volunteer librarians will visit neighborhoods equipped with book bags for engaging discussions featuring several beloved children's authors such as Ann M . Robbins , Diane Trela , Mary Pope Osborne , Beverly Cleary , Eric Carle , Chris Van Allsburg among others ! Your local library will host a pair of enjoyable free events this autumn . On Saturday , November 29th between 10 AM and 11 AM , come enjoy refreshments like cookies and coffee while listening to a presentation of "Where the Kids Play . " Author- illustrator Ann M . Robbins will be present for a live reading of her new work that introduces characters like Miss Mabel along with a couple of feline companions known as Fat and Happy . If you 'd rather focus on literature than snacks , a session will also feature a read - aloud of last season 's popular choice for the autumn book club : "Where the Bats Don ' t ...

## M.2. Examples of text degeneration

---
**Example1**

In a highly anticipated matchup between two exceptional pitchers, Johan Santana (6–4, 3.16 ERA) faced off against Curt Schilling (9–2, 3.81 ERA) – a contest where Schilling had previously earned a 2004 Cy Young Award victory over a runner-up performance by Santana.

Schilling reflected on their encounter by stating, "I knew right away he had what we needed." He believed a single error would be enough for his team to secure a win.

For eight impressive innings, Santana demonstrated remarkable prowess by racking up an impressive total of 13 strikeouts – a figure only surpassed by a left-handed pitcher once before – when Tom Seaver achieved this feat in 1967.  though there were two other pitchers who had done so before him: Tom Seaver achieved this feat before this game but also a notable left-handed pitcher who achieved this feat before him: Sandy Koufax achieved this feat before this game; Tom Seaver actually achieved this feat before this game; Sandy Koufax achieved this feat before this game; Tom Seaver achieved this feat before this game; Sandy Koufax achieved this feat before this game; Tom Seaver achieved this feat before this game; Sandy Koufax achieved this feat before this game; Tom Seaver achieved this feat before this game; Sandy Koufax achieved this feat before this game; Tom Seaver achieved this feat before this game; Sandy Koufax achieved this feat before this game; Tom Seaver achieved this feat before this game; Sandy Koufax achieved this feat before this game; Tom Seaver achieved this feat before this game; Sandy Koufax achieved this feat before this game, ...

---
**Example2**

Bearing testament to this self-awareness, Mike's recounting of an experience involving none other than the legendary Viv Richards, often referred to as 'the Original King', showcases vividly how humbled he remained, despite being on opposite sides of an intense rivalry, especially evident within the dedicated chapter devoted to their storied encounter, where Mike pens about being, to put it mildly, utterly perplexed by how Mr Richards chose to treat him, revealing Mike's profound recognition and acceptance of his own limitations on the field, particularly amidst such high-caliber competition, like facing off against one of history's greatest batsmen, who undoubtedly left an indelible mark on Mike's memory, even to this day, an experience vividly captured within those pages, offering valuable insights into Mike's candid account, one deeply rooted within genuine humility, self-awareness, an extraordinary capacity to reflect, coupled by an all-consuming passion to explore, understand, analyze, learn, grow, an essential, enduring component, now woven into Mike's legacy. cherished, loved by all, forever, truly, an inspiration to many, today, now, always, cherished, loved by all, forever, truly, an inspiration to many, today, now, always, cherished, loved by all, forever, truly, an inspiration, an icon, cherished, loved, forever, truly, an inspiration, an icon, cherished, loved, forever, truly, an inspiration, an icon, cherished, loved, cherished, loved by all, forever, truly, an inspiration to many, today, now, always, cherished, loved by all, cherished, loved by all, forever, truly, an inspiration to many, today, now, always, cherished, loved by all, cherished, loved by all, forever, truly, an inspiration to many, today, now, always, cherished, loved by all, cherished, loved by all, forever, truly, an inspiration, ...

---

## M.3. Example of a stiff text in GPT-4o-mini

**Watermarked Text1 by KGW**

Harry Connolly's Twenty Palaces series is one of my favorite new storylines in science fiction. After a several year hiatus Harry has brought back Raynor with a new twist on the concept in 20 Palaces of Niven. In this sequel to Thrive there is something even better than the original Twenty Palaces (I mean, duh) - a war. The premise for this story takes the format of earlier books - the protagonist goes to visit each successive palaces of the King, and learns how to interact with inhabitants in the culture, and so on - but introduces an element of fantasy: that of humans in space that have been rendered extinct by a war-based plague that ended the universe (that is, until this novel came along). This raises an interesting question - are dragons actually extinct? And if not, how do you rebuild, and on what do they rebuild? The answer isn't really that simple - but Connolly does an amazing job of laying it out. With this book, he doesn't get bogged down with "what happens after the end of the end of everything?" and "what happens to all of humanity".

**Attacked Text1 by BIRA (PPL=25.0)**

Harry Connolly's Twenty Palaces saga ranks among my favorite recent narratives within sci-fi. Following a multi-year break, he has reintroduced Raynor along with a fresh perspective through his work titled *20 Palaces of Niven*. This continuation to *Thrive* offers a storyline that's superior to its predecessor—naturally—due to its inclusion of a conflict. The narrative structure mirrors previous installments where the main character visits various royal residences while learning to engage with their cultures. However, it adds a fantastical element involving humanity's extinction caused by a warfare-induced pandemic which obliterated existence—until this tale emerged. This prompts a compelling inquiry—are dragons truly gone? If not, what does rebuilding entail? The solution isn't straightforward; nonetheless, Connolly excels at presenting it. He skillfully avoids getting mired down by questions like "what follows when all seems lost?" or "what becomes of humanity?"

**Watermarked Text2 by KGW**

In the summer of 1993, I had one of those experiences that stay with me for the rest of my life. I spent 10 weeks in China volunteering for an orphanage, in one particularly sad season of the children's lives. And, during that time, I had to walk to take pictures of the kids in their houses, because they could never leave their houses on their own to go find the photos somewhere nearby. I was alone each time and walking to and from the children's houses, I could see and hear all sorts of things: the way that the children fidgeted when there were other children around, the sadness and sorrowful looks that lit up their faces. I had never met any of the young children of the orphanage before I went. How did these little babies get there? How did I get there to hold my arms around them? And why are they sad? If they wanted, could I change their situation? If any of the children had been adopted, would other parents love them as much? As I walked the paths and found a way from their houses into one of the...

**Attacked Text2 by BIRA (PPL=19.13)**

In the summertime of 1993, I experienced something unforgettable that has remained with me throughout my life. For ten weeks, I volunteered at an orphanage in China during what was especially difficult times for those kids. During this period, it was necessary for me to stroll over to capture photographs of them at their homes since none were able to leave their residences independently to seek out their images elsewhere close by. Each time was solitary as I walked back and forth; along those routes, I observed various things—the mannerisms with which the kids would squirm around others present as well as their faces illuminated by expressions filled with grief and melancholy. Prior to this experience at the orphanage, none of those small kids were familiar figures to me. What circumstances led those tiny infants here? What brought me here so as to embrace them? What is causing their unhappiness? Could their circumstances be altered if given half a chance? Would adoptive parents cherish them just as much if some were taken home? As I traversed those pathways leading away from their homes toward another location...

# N. Detailed Experimental Results

## N.1. Detailed experimental results for dynamic threshold

*Table 17.* Best F1 Score (%) across different models and watermarking algorithms.

| Model \ Watermark | KGW | Unigram | UPV | EWD | DIP | SIR | EXP |
|---|---|---|---|---|---|---|---|
| Vanilla (Llama-3.1-8B) | 0.863 | 0.9 | 0.907 | 0.831 | 0.67 | 0.893 | 0.666 |
| Vanilla (Llama-3.1-70B) | 0.895 | 0.912 | 0.939 | 0.855 | 0.671 | 0.919 | 0.666 |
| Vanilla (GPT-4o-mini) | 0.955 | 0.969 | 0.97 | 0.951 | 0.71 | 0.964 | 0.666 |
| DIPPER-1 | 0.84 | 0.918 | 0.908 | 0.871 | 0.668 | 0.869 | 0.666 |
| DIPPER-2 | 0.81 | 0.89 | 0.867 | 0.81 | 0.667 | 0.824 | 0.666 |
| SIRA (Llama-3.1-8B) | 0.727 | 0.78 | 0.806 | 0.715 | 0.667 | 0.803 | 0.666 |
| SIRA (Llama-3.1-70B) | 0.783 | 0.845 | 0.828 | 0.73 | 0.668 | 0.84 | 0.666 |
| SIRA (GPT-4o-mini) | 0.781 | 0.875 | 0.837 | 0.799 | 0.667 | 0.872 | 0.666 |
| BIRA (Llama-3.1-8B) | 0.723 | 0.666 | 0.667 | 0.683 | 0.668 | 0.666 | 0.666 |
| BIRA (Llama-3.1-70B) | 0.723 | 0.667 | 0.666 | 0.682 | 0.667 | 0.667 | 0.666 |
| BIRA (GPT-4o-mini) | 0.751 | 0.668 | 0.668 | 0.69 | 0.667 | 0.667 | 0.666 |

*Table 18.* TPR under 1% FPR (%) across different models and watermarking algorithms.

| Model \ Watermark | KGW | Unigram | UPV | EWD | DIP | SIR | EXP |
|---|---|---|---|---|---|---|---|
| Vanilla (Llama-3.1-8B) | 0.47 | 0.612 | 0.572 | 0.424 | 0.022 | 0.638 | 0.006 |
| Vanilla (Llama-3.1-70B) | 0.566 | 0.722 | 0.708 | 0.524 | 0.052 | 0.748 | 0.0 |
| Vanilla (GPT-4o-mini) | 0.842 | 0.932 | 0.864 | 0.864 | 0.144 | 0.902 | 0.006 |
| DIPPER-1 | 0.376 | 0.762 | 0.646 | 0.518 | 0.032 | 0.614 | 0.008 |
| DIPPER-2 | 0.226 | 0.68 | 0.5 | 0.316 | 0.02 | 0.424 | 0.004 |
| SIRA (Llama-3.1-8B) | 0.126 | 0.242 | 0.266 | 0.094 | 0.014 | 0.438 | 0.012 |
| SIRA (Llama-3.1-70B) | 0.224 | 0.438 | 0.344 | 0.174 | 0.014 | 0.552 | 0.004 |
| SIRA (GPT-4o-mini) | 0.21 | 0.498 | 0.358 | 0.266 | 0.024 | 0.574 | 0.002 |
| BIRA (Llama-3.1-8B) | 0.084 | 0.038 | 0.036 | 0.042 | 0.018 | 0.012 | 0.014 |
| BIRA (Llama-3.1-70B) | 0.092 | 0.052 | 0.054 | 0.078 | 0.014 | 0.042 | 0.012 |
| BIRA (GPT-4o-mini) | 0.12 | 0.022 | 0.04 | 0.05 | 0.012 | 0.024 | 0.024 |

*Table 19.* TPR under 10% FPR (%) across different models and watermarking algorithms.

| Model \ Watermark | KGW | Unigram | UPV | EWD | DIP | SIR | EXP |
|---|---|---|---|---|---|---|---|
| Vanilla (Llama-3.1-8B) | 0.772 | 0.878 | 0.894 | 0.71 | 0.186 | 0.882 | 0.042 |
| Vanilla (Llama-3.1-70B) | 0.854 | 0.912 | 0.96 | 0.804 | 0.204 | 0.928 | 0.002 |
| Vanilla (GPT-4o-mini) | 0.98 | 0.994 | 0.988 | 0.968 | 0.404 | 0.974 | 0.008 |
| DIPPER-1 | 0.714 | 0.918 | 0.902 | 0.812 | 0.158 | 0.832 | 0.026 |
| DIPPER-2 | 0.6 | 0.86 | 0.808 | 0.656 | 0.128 | 0.728 | 0.026 |
| SIRA (Llama-3.1-8B) | 0.406 | 0.576 | 0.602 | 0.33 | 0.108 | 0.7 | 0.06 |
| SIRA (Llama-3.1-70B) | 0.548 | 0.73 | 0.698 | 0.466 | 0.114 | 0.766 | 0.034 |
| SIRA (GPT-4o-mini) | 0.566 | 0.82 | 0.712 | 0.588 | 0.162 | 0.824 | 0.026 |
| BIRA (Llama-3.1-8B) | 0.356 | 0.114 | 0.254 | 0.212 | 0.1 | 0.114 | 0.108 |
| BIRA (Llama-3.1-70B) | 0.346 | 0.152 | 0.26 | 0.278 | 0.102 | 0.18 | 0.092 |
| BIRA (GPT-4o-mini) | 0.428 | 0.116 | 0.218 | 0.236 | 0.116 | 0.112 | 0.092 |

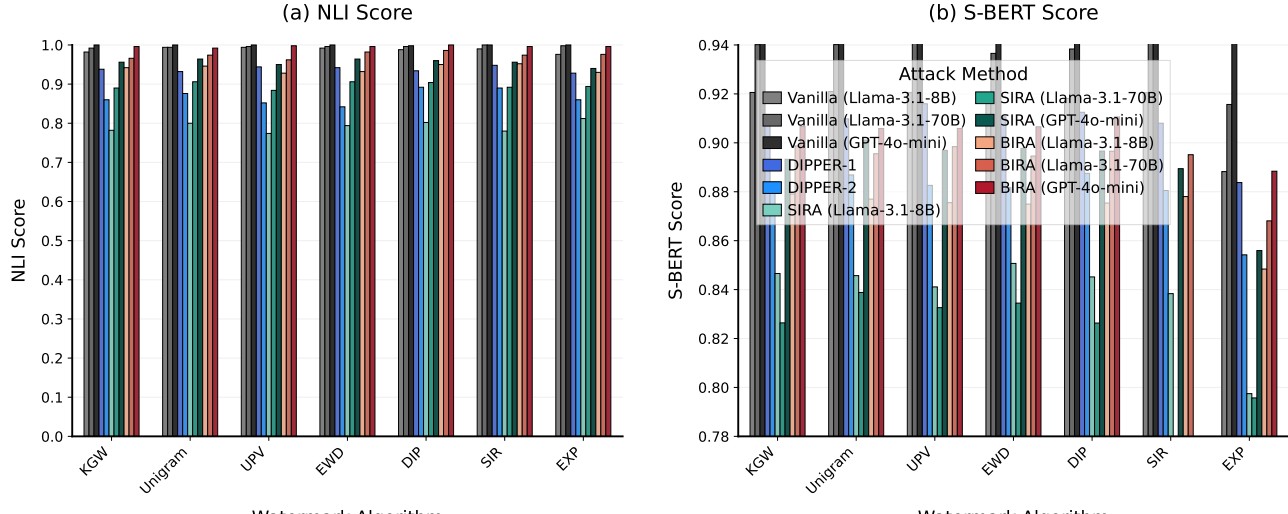

*Figure 15.* Comparison of text quality across different attacks for various watermarking methods, evaluated by NLI score (↑) and S-BERT score (↑). Our method is comparable to or outperforms other baselines on both metrics. Following (Cheng et al., 2025), we evaluate attacks on S-BERT score. However, we observe that the S-BERT score often fails to capture factual accuracy and fine-grained meaning, sometimes assigning high scores despite factual errors and low scores even when the original meaning is preserved, likely because heavily paraphrased text is less familiar to the model.

## N.2. Detailed experimental results of text quality evaluation

*Table 20.* **LLM Judgement Score** (↑) across different models and watermarking algorithms.

| Model / Watermark | KGW | Unigram | UPV | EWD | DIP | SIR | EXP | Avg Score |
|---|---|---|---|---|---|---|---|---|
| Vanilla (Llama-3.1-8B) | 4.774 | 4.728 | 4.778 | 4.786 | 4.776 | 4.76 | 4.644 | 4.749 |
| Vanilla (Llama-3.1-70B) | 4.906 | 4.914 | 4.93 | 4.852 | 4.918 | 4.894 | 4.874 | 4.898 |
| Vanilla (GPT-4o-mini) | 4.986 | 4.96 | 4.99 | 4.984 | 4.98 | 4.974 | 4.984 | 4.98 |
| DIPPER-1 | 3.398 | 3.504 | 3.636 | 3.464 | 3.446 | 3.55 | 3.088 | 3.441 |
| DIPPER-2 | 3.076 | 3.092 | 3.204 | 3.034 | 3.016 | 3.122 | 2.734 | 3.04 |
| SIRA (Llama-3.1-8B) | 3.356 | 3.34 | 3.356 | 3.348 | 3.364 | 3.25 | 3.142 | 3.308 |
| SIRA (Llama-3.1-70B) | 3.708 | 3.72 | 3.744 | 3.724 | 3.72 | 3.654 | 3.67 | 3.706 |
| SIRA (GPT-4o-mini) | 4.418 | 4.49 | 4.496 | 4.488 | 4.506 | 4.334 | 4.292 | 4.432 |
| BIRA (Llama-3.1-8B) | 4.212 | 4.212 | 4.306 | 4.234 | 4.296 | 4.24 | 4.084 | 4.226 |
| BIRA (Llama-3.1-70B) | 4.524 | 4.454 | 4.528 | 4.484 | 4.544 | 4.42 | 4.342 | 4.471 |
| BIRA (GPT-4o-mini) | 4.722 | 4.688 | 4.736 | 4.728 | 4.778 | 4.75 | 4.708 | 4.73 |

*Table 21.* **Self-BLEU Score** (↓) across different models and watermarking algorithms.

| Model / Watermark | KGW | Unigram | UPV | EWD | DIP | SIR | EXP | Avg Score |
|---|---|---|---|---|---|---|---|---|
| Vanilla (Llama-3.1-8B) | 0.247 | 0.25 | 0.252 | 0.248 | 0.265 | 0.25 | 0.224 | 0.248 |
| Vanilla (Llama-3.1-70B) | 0.282 | 0.281 | 0.295 | 0.284 | 0.298 | 0.287 | 0.268 | 0.285 |
| Vanilla (GPT-4o-mini) | 0.422 | 0.407 | 0.436 | 0.421 | 0.435 | 0.409 | 0.414 | 0.42 |
| DIPPER-1 | 0.26 | 0.259 | 0.291 | 0.267 | 0.28 | 0.255 | 0.233 | 0.263 |
| DIPPER-2 | 0.196 | 0.199 | 0.205 | 0.196 | 0.204 | 0.19 | 0.176 | 0.195 |
| SIRA (Llama-3.1-8B) | 0.108 | 0.111 | 0.108 | 0.105 | 0.119 | 0.107 | 0.102 | 0.109 |
| SIRA (Llama-3.1-70B) | 0.151 | 0.153 | 0.149 | 0.153 | 0.159 | 0.155 | 0.142 | 0.151 |
| SIRA (GPT-4o-mini) | 0.173 | 0.174 | 0.176 | 0.181 | 0.192 | 0.173 | 0.16 | 0.176 |
| BIRA (Llama-3.1-8B) | 0.071 | 0.07 | 0.068 | 0.069 | 0.077 | 0.066 | 0.067 | 0.07 |
| BIRA (Llama-3.1-70B) | 0.089 | 0.085 | 0.084 | 0.088 | 0.091 | 0.08 | 0.082 | 0.086 |
| BIRA (GPT-4o-mini) | 0.079 | 0.076 | 0.076 | 0.078 | 0.088 | 0.075 | 0.08 | 0.079 |

*Table 22.* **Perplexity** (↓) across different models and watermarking algorithms.

| Model / Watermark | KGW | Unigram | UPV | EWD | DIP | SIR | EXP | Avg Score |
|---|---|---|---|---|---|---|---|---|
| Vanilla (Llama-3.1-8B) | 8.931 | 8.783 | 7.941 | 8.814 | 8.375 | 9.163 | 10.64 | 8.95 |
| Vanilla (Llama-3.1-70B) | 9.315 | 9.167 | 8.078 | 9.236 | 8.564 | 9.427 | 11.434 | 9.317 |
| Vanilla (GPT-4o-mini) | 11.272 | 11.202 | 9.515 | 11.19 | 10.224 | 11.481 | 13.636 | 11.217 |
| DIPPER-1 | 10.953 | 10.743 | 9.217 | 10.719 | 10.506 | 11.199 | 13.659 | 10.999 |
| DIPPER-2 | 11.371 | 10.973 | 9.769 | 10.956 | 10.886 | 11.466 | 13.831 | 11.322 |
| SIRA (Llama-3.1-8B) | 9.099 | 9.275 | 9.155 | 8.888 | 9.218 | 9.984 | 10.51 | 9.447 |
| SIRA (Llama-3.1-70B) | 9.659 | 9.494 | 8.748 | 9.381 | 9.361 | 9.860 | 11.314 | 9.66 |
| SIRA (GPT-4o-mini) | 9.39 | 9.139 | 8.515 | 9.279 | 8.906 | 9.453 | 11.528 | 9.459 |
| BIRA (Llama-3.1-8B) | 10.586 | 10.458 | 9.864 | 10.54 | 10.33 | 10.189 | 11.813 | 10.54 |
| BIRA (Llama-3.1-70B) | 12.367 | 11.864 | 11.463 | 12.065 | 11.585 | 11.539 | 13.632 | 12.074 |
| BIRA (GPT-4o-mini) | 15.99 | 15.725 | 14.434 | 15.765 | 14.711 | 15.067 | 17.106 | 15.543 |

*Table 23.* **NLI Score** (↑) across different models and watermarking algorithms.

| Model / Watermark | KGW | Unigram | UPV | EWD | DIP | SIR | EXP | Avg Score |
|---|---|---|---|---|---|---|---|---|
| Vanilla (Llama-3.1-8B) | 0.982 | 0.994 | 0.994 | 0.992 | 0.988 | 0.99 | 0.976 | 0.988 |
| Vanilla (Llama-3.1-70B) | 0.992 | 0.994 | 0.996 | 0.996 | 0.996 | 1.0 | 0.998 | 0.996 |
| Vanilla (GPT-4o-mini) | 1.0 | 1.0 | 1.0 | 1.0 | 0.998 | 1.0 | 1.0 | 1.0 |
| DIPPER-1 | 0.938 | 0.932 | 0.944 | 0.942 | 0.934 | 0.948 | 0.928 | 0.938 |
| DIPPER-2 | 0.86 | 0.876 | 0.852 | 0.842 | 0.892 | 0.89 | 0.86 | 0.867 |
| SIRA (Llama-3.1-8B) | 0.782 | 0.8 | 0.774 | 0.794 | 0.802 | 0.78 | 0.812 | 0.792 |
| SIRA (Llama-3.1-70B) | 0.89 | 0.906 | 0.884 | 0.906 | 0.904 | 0.892 | 0.894 | 0.897 |
| SIRA (GPT-4o-mini) | 0.956 | 0.964 | 0.95 | 0.964 | 0.96 | 0.956 | 0.94 | 0.956 |
| BIRA (Llama-3.1-8B) | 0.942 | 0.946 | 0.928 | 0.932 | 0.95 | 0.952 | 0.93 | 0.94 |
| BIRA (Llama-3.1-70B) | 0.966 | 0.974 | 0.962 | 0.982 | 0.986 | 0.974 | 0.976 | 0.974 |
| BIRA (GPT-4o-mini) | 0.996 | 0.992 | 0.998 | 0.996 | 1.0 | 0.996 | 0.996 | 0.996 |

*Table 24.* **S-BERT Score** (↑) across different models and watermarking algorithms.

| Watermark
Model | KGW | Unigram | UPV | EWD | DIP | SIR | EXP | Avg Score |
|---|---|---|---|---|---|---|---|---|
| Vanilla (Llama-3.1-8B) | 0.921 | 0.921 | 0.920 | 0.918 | 0.918 | 0.920 | 0.888 | 0.915 |
| Vanilla (Llama-3.1-70B) | 0.940 | 0.940 | 0.944 | 0.937 | 0.938 | 0.941 | 0.916 | 0.936 |
| Vanilla (GPT-4o-mini) | 0.965 | 0.964 | 0.965 | 0.963 | 0.966 | 0.963 | 0.955 | 0.963 |
| DIPPER-1 | 0.908 | 0.910 | 0.916 | 0.911 | 0.912 | 0.908 | 0.884 | 0.907 |
| DIPPER-2 | 0.883 | 0.887 | 0.883 | 0.882 | 0.888 | 0.881 | 0.854 | 0.879 |
| SIRA (Llama-3.1-8B) | 0.847 | 0.846 | 0.841 | 0.851 | 0.845 | 0.838 | 0.797 | 0.838 |
| SIRA (Llama-3.1-70B) | 0.826 | 0.839 | 0.833 | 0.834 | 0.826 | 0.828 | 0.796 | 0.826 |
| SIRA (GPT-4o-mini) | 0.893 | 0.901 | 0.897 | 0.898 | 0.897 | 0.889 | 0.856 | 0.890 |
| BIRA (Llama-3.1-8B) | 0.879 | 0.877 | 0.876 | 0.875 | 0.875 | 0.878 | 0.848 | 0.873 |
| BIRA (Llama-3.1-70B) | 0.899 | 0.896 | 0.898 | 0.895 | 0.897 | 0.895 | 0.868 | 0.892 |
| BIRA (GPT-4o-mini) | 0.907 | 0.906 | 0.906 | 0.907 | 0.911 | 0.908 | 0.888 | 0.905 |

