# OpenReview forum: "LLM Watermark Evasion via Bias Inversion"
_ICML.cc/2026/Conference — ICML 2026 regular_

### Official Review · Reviewer_h4Ch · 2026-03-06

**Soundness:** 2
**Presentation:** 3
**Significance:** 3
**Originality:** 3
**Overall Recommendation:** 4
**Confidence:** 5

**Summary:**

This paper studies watermark evasion and removal for LLM generated text. It proposes a black box rewriting attack called BIRA, short for Bias Inversion Rewriting Attack. The attack does not require the secret key and does not rely on repeated queries to the victim model. The main idea is based on a simple prior that watermark signals are more likely to show up at high entropy positions, meaning tokens with high surprisal. The method uses a public language model to compute self information on the input watermarked text and builds a proxy suppression token set. During rewriting, it applies a negative logit bias to tokens in this set so they are less likely to appear again. This indirectly weakens the green token statistics that many detectors rely on. The paper also adds a degeneration check and an adaptive bias schedule to reduce collapse during generation. Experiments cover several widely used token level watermarking schemes and include an additional study on a sentence level watermark called SimMark. Overall, the results suggest the attack can reach a very high success rate while keeping semantics reasonably well.

**Compliance With Llm Reviewing Policy:**

Affirmed.

**Final Justification:**

Thank you to the authors for the detailed response to my comments. I have also reviewed the other reviewers’ comments and the authors’ responses to them. Based on the above, I have made my final decision as 4: Weak Accept.

**Key Questions For Authors:**

(1) The proxy set $G_b$ is built from surprisal computed by a public language model. Please provide a systematic ablation where you keep the watermarked text and the rewriting model fixed, and only vary the model used to compute surprisal, using different architectures, different tokenizers, and different model sizes. It would be helpful to report how the stability of $G_b$, measured by overlap, as well as ASR, semantic preservation, and degeneration rate change under these settings. With the current presentation, it is hard to judge how robust the approach is under realistic heterogeneity.

(2) On the dependence boundary of the high entropy signal assumption: If a watermark is deliberately designed to distribute its signal more evenly across positions, to place more signal on lower entropy tokens, or to move the signal toward sentence level or semantic level features, can BIRA still maintain a high ASR without noticeably sacrificing semantics. A stronger theoretical or empirical analysis would help clarify the failure conditions and the associated cost, such as a quality degradation curve as $\beta$ becomes more negative.

(3) On the adequacy of comparisons against robust watermarks: The main experiments focus on token level watermarks and only include limited additional validation on a small number of sentence level schemes. Since the paper is framed around evasion and removal, it would strengthen the evaluation to include multiple watermark designs that explicitly aim to be robust to paraphrasing and scrubbing, covering at least one sentence level method, one semantic level method, and one redundancy based design. Under comparable semantic constraints, please report detector performance such as TPR at a fixed FPR, along with the attack cost.

(4) On details of proxy set construction: The current method selects high surprisal tokens by a percentile q and aggregates them into a token id set. Please clarify whether this proxy is defined at the token type level as token ids or at the position level. If the same token appears in different contexts, is it always suppressed indiscriminately. How controllable is the impact on semantic preservation. Have you tried position level or local window level proxy variants as alternatives.

(5) On the impact of degeneration detection and restart strategy: The distinct one gram ratio threshold $\rho$, the window size $h$, and the maximum number of restarts $R$ may have a large impact on both success and quality. Please provide a sensitivity analysis and clarify whether these parameters need to be retuned across different tasks or different output lengths.

**Limitations:**

First, it relies heavily on the prior that watermark traces tend to concentrate on high entropy tokens, so if a watermark is designed to spread its signal more evenly, place more signal on lower entropy positions, or move the signal to sentence level or semantic level features, the proxy set may become weakly correlated with the true watermark signal and the attack may need a much stronger negative bias, which can hurt meaning and fluency. Second, the approach is sensitive to heterogeneous models and tokenizer alignment, since the proxy is selected in the token space of a public surprisal model while bias injection happens in the token space of the rewriting model, and mismatch can make the suppression noisy or overly coarse. Third, because BIRA aggregates high surprisal observations into a global token id set and suppresses those tokens throughout decoding, it can be overly blunt by discouraging useful words even when they are not carrying watermark traces, increasing rewrite pressure and raising the risk of semantic drift or degeneration.

**Strengths And Weaknesses:**

1. Soundness

The paper connects its theory and method in a fairly clean way by treating common watermark detectors as monotonic tests on the fraction of green tokens, then arguing that it only takes a small average drop in the probability of sampling green tokens to make the detection probability fall exponentially, which gives a solid motivation for the bias inversion strategy. On the practical side, the algorithm is designed with usability in mind, since it recognizes that a strong negative bias can trigger degeneration, so it monitors degeneration with a distinct one gram ratio style criterion and adapts the bias strength on the fly to keep generation stable.

Weakness:

(1) A key weakness is that the proxy set design is still more heuristic than principled. The method builds the proxy suppression set by taking tokens with high surprisal under a public language model, based on the intuition that watermark effects tend to surface at high entropy positions. This makes sense for many current token level schemes, but the paper does not really pin down how strongly this proxy correlates with the true dynamic green set induced by the secret key and the local context. More importantly, it does not explore the failure modes where watermark designers deliberately spread the signal more evenly across positions, shift the signal toward lower entropy tokens, or move the watermark to higher level features such as sentence level or structural patterns. In these settings, the proxy set could become weakly related to the actual watermark traces, and the attack might need a much stronger bias to succeed, which would likely increase degeneration and hurt semantic fidelity. A more systematic analysis of this boundary would make the claims feel more robust.

(2) Another limitation is the reliance on cross model and cross tokenizer alignment, which is a real issue in the stated black box threat model. In practice, the model used to compute surprisal, the rewriting model that applies logit biases, the unknown model that produced the watermarked text, and the detector tokenizer may all differ. When tokenization is not aligned, the proxy tokens selected under one tokenizer do not map cleanly to the token ids of another model, so the bias injection becomes noisy or overly coarse. The paper does mention an engineering workaround in the GPT setting by converting selected tokens to text and retokenizing them before applying token level bias, but it does not provide a careful ablation that quantifies how much this mismatch changes attack success rate, semantic preservation, and degeneration frequency. A controlled study that varies the surprisal model and tokenizer while holding the rest fixed would help clarify how stable the approach is under realistic heterogeneity.

(3) The evaluation mainly targets standard token level watermarks. Since the paper is framed as a general watermark evasion and removal attack, it would strengthen the soundness to include comparisons against watermark designs that explicitly aim to be robust to paraphrasing and scrubbing. Otherwise, the current results may primarily reflect the vulnerability of green token bias style detectors rather than a broader conclusion about robust watermarking.

2. Presentation

The paper is generally easy to follow, with a clear background to theory to algorithm to experiments flow that helps readers understand why proxy suppression and bias inversion are introduced, and the evaluation is fairly complete since it reports attack success together with detector side metrics such as TPR and F1 under fixed FPR, as well as semantic and fluency related measures, which makes the tradeoff between evasion strength and meaning preservation straightforward to interpret.

Weakness:

(1) It would help to unify the notation and clearly state the role of each symbol when it first appears. For instance, at Line 108 the paper says where S is a similarity measure between two texts, while at Line 166 it lists Input: System prompt S. Reusing S for different meanings is confusing and should be avoided.

(2) The discussion of hyperparameters and degeneration control feels scattered. Parameters such as q, beta, h, rho, lr, and R appear in the pseudocode, but the main text does not give a consolidated explanation of what each one does or how the default settings are chosen. This makes the method harder to reproduce than it needs to be.

3. Significance

The paper presents a strong black box evasion approach against watermarking schemes that rely on green token bias and simple statistical tests, and it highlights a clear takeaway for the community: if a watermark is mainly encoded as token level statistical signals and those signals concentrate on high entropy positions, then they can be exploited. The attack also shows reasonable transfer across different rewriting models, including LLMs of different sizes, which makes the threat model feel realistic rather than purely academic.

Weakness:

At the moment, the results read more like an upper bound stress test on mainstream token level statistical watermarks, rather than a general refutation of watermarking as a whole. Given that more and more watermark designs are explicitly targeting robustness to rewriting, the limited coverage of sentence level robust watermarks is not enough to support a broad claim of universal removal capability.

4. Originality

The paper combines self information based high entropy token localization with logit bias inversion during decoding. Compared with explicit mask and rewrite attacks, this gives a more continuous and controllable strategy at the probability level, and it also includes a practical degeneration control mechanism. On the theory side, it abstracts common detectors as monotonic functions and derives an exponential decay bound, which provides an interpretable mathematical motivation for the attack.

Weakness:

The key prior and overall direction of using self information to identify high entropy tokens is closely related to existing work such as SIRA. The main contribution here seems to be more about how to use a proxy set for decoding time bias suppression and how to make the method stable in practice. It would help if the authors more clearly separate their core differences from the closest baselines and explain what is truly non substitutable.

---

> ### Author Rebuttal · Authors · 2026-03-30
>
> We sincerely appreciate your taking the time to review. We have carefully considered your thoughtful comments and provide detailed responses below.
>
> > W1-1. Correlation between proxy and true dynamic green set
>
> The proxy set is designed to suppress the realized watermark trace under the context of the watermarked text during rewriting, rather than to recover the dynamic green set across all contexts. We therefore assess this via a hit-rate analysis on KGW-watermarked text, measuring whether entropy-selected tokens belong to the dynamic green set at their original contexts. Using KGW text generated by Llama-3.2-3B and Llama-3.1-8B as rewriters, which share the same tokenizer, we obtain a hit rate of 0.819 at a selection ratio of 0.2, compared with 0.713 for random selection. This is also consistent with the ASR ablation in Fig. 6, where self-information-guided selection consistently outperforms the random selection.
>
> > W1-1&Q2. Failure modes: (1) signal shift toward lower entropy tokens (2) spread the signal evenly across positions (3) sentence-level
>
> For (1), shifting more signal to low-entropy tokens is possible in principle, but typically requires stronger biases on constrained token choices, which substantially harm text quality and limit practical use. This is also why watermarking is difficult in constrained domains, e.g., code generation. For (2), spreading the signal evenly across positions still relies on having enough token entropy to preserve quality, and at such positions, BIRA can still suppress the signal by steering generation toward alternative tokens. For (3), sentence-level watermarking depends on consistent sentence segmentation between generation and detection, whereas BIRA performs document-level paraphrasing that may merge or split sentences, weakening this assumption.
>
> > W1-2&Q1. Cross model and tokenizer alignment
>
> In our setup, the surprisal model is the same as the rewriting model, matching the setting where an adversary runs an LLM locally. The only exception is GPT-4o-mini, which does not expose token-level logits. Fixing the SIR-watermarked text and GPT rewriter while varying the surprisal model across (1) Llama-3.2-3B, (2) Llama-3.1-8B, (3) Qwen3-8B, and (4) Phi-4-mini yields stable performance (Table below). The Jaccard similarity between (1) and (2), which contain the same token IDs, is 0.88, indicating substantial overlap.
>
> |Metric|Avg|
> |-|-|
> |TPR@FPR=1%|0.019±0.005|
> |LLM-judge|4.72±0.03|
> |# iterations|1.02±0.01|
>
> > W1-3&W3&Q3. Extended evaluation: (1) semantic-level, (2) sentence-level, and (3) redundancy
>
> For (1), we already include SIR, a semantic token-level watermark, and for (2), SimMark. Below, we further evaluate PMARK for (2) and WatME for (3) using the official code and settings from the papers. Due to space limits, we report only results for our method/SIRA below.
>
> |Setting|TPR@FPR=1%|LLM-judge|
> |-|-|-|
> |PMARK(2)|0.04/0.03|4.35/3.42|
> |WatME(3)|0.015/0.015|4.63/3.64|
>
> Attack cost is reported in Appendix D.
>
> > W2-1. Unified notation
>
> We will revise these points.
>
> > W2-2&Q5. Hyperparameters
>
> The beta selection strategy and lr are described in line 225 (L), and q is fixed at 50 as a simple midpoint choice. Table 2 provides ablations for beta and q, and Appendix C discusses the hyperparameters for degeneration detection. We set R=20. While beta varies by model under this strategy (line 239, R), all other hyperparameters are fixed across models, datasets (Appendix E), and output lengths (below). We will consolidate these details and upload the code.
>
> > W4. Difference between BIRA and SIRA
>
> The core difference is that BIRA is directly motivated by our theorem and offers a more controllable approach at the decoding step with an adaptive bias strategy. This enables both high semantic fidelity and high attack success rates, a balance largely overlooked in SIRA. This advantage is further highlighted in the long-text experiments on the 600-token SIR watermark (the table in our response to the first reviewer, kQwM), where SIRA largely fails to preserve semantic fidelity.
>
> > Q4. Proxy set
>
> We compute token surprisal at each position, collect the IDs of high-surprisal tokens, and use them to construct a proxy set. Although suppression is applied globally, it is implemented as a soft negative bias rather than a hard constraint. As a result, actual token generation remains context-dependent, so even tokens in the proxy set can still be generated when they are strongly favored in context. This is consistent with our strong semantic preservation results.
>
> Local or position-level proxies are interesting alternatives, but they are not straightforward in our setting because BIRA may rewrite sentence structure by merging or splitting sentences. Possible mitigations would be to apply BIRA chunk-wise and recombine the outputs, or to make the bias strength itself more context-adaptive. Exploring such alternatives is a promising direction for future work to further improve text quality.

---

> > ### Author Rebuttal · Reviewer_h4Ch · 2026-04-03
> >
> > Thank you to the authors for the detailed rebuttal and additional experiments. I acknowledge that the response has resolved most of the concerns I raised. On Q2: The rebuttal provides only theoretical arguments, not the empirical analysis I suggested. I recommend adding controlled experiments (e.g., watermark variants with different entropy distributions) to show how ASR and semantic quality degrade under stronger bias or shifted signals. This would strengthen the paper's claims.
> >
> > For all, I maintain my original score.

---

> > > ### Author Response · Authors · 2026-04-05
> > >
> > > We appreciate your constructive comment. We provide an additional empirical analysis of watermark variants with different entropy distributions under the same setup as Section 5.1, using Llama 3.1-8B as the rewriting model and focusing on SIR, the most robust watermark in our experiments.
> > >
> > > > (1) Shifting signal toward lower-entropy tokens.
> > >
> > > We implement a low-entropy variant by applying the watermark bias $\gamma > 0$ only at low-entropy token positions. Specifically, we calculate token entropy from 500 natural-text samples, compute the 30th- and 50th-percentile entropy thresholds for each sentence, average them across samples, and apply and detect the watermark only on tokens below the selected threshold.
> > >
> > > The first table reports watermark-only performance under the 30th/50th-percentile thresholds, together with the standard SIR for reference. Relative to standard SIR with $\gamma = 1.0$, achieving reliable detectability in this setting requires a substantially larger $\gamma$. This is because low-entropy positions allow less perturbation, and restricting watermarking to this subset reduces the total signal available to the detector. At the same time, increasing $\gamma$ sharply raises PPL and noticeably degrades text quality. When $\gamma \geq 2.0$, we observe frequent grammatical errors, unusual spelling, and repetitive phrases because the stronger bias forces changes to highly deterministic tokens. Thus, shifting the watermark signal toward low-entropy positions does not provide a practical defense. Instead, it leads to a substantially worse detectability-quality tradeoff.
> > >
> > > |Setting|TPR@FPR=1%|TPR@FPR=10%|PPL|
> > > |-|-|-|-|
> > > |SIR|0.998|1.000|11.95|
> > > |Low-entropy,$\gamma=1.0$|0.098/0.596|0.368/0.884|9.87/10.64|
> > > |Low-entropy,$\gamma=2.0$|0.612/0.988|0.896/1.000|12.07/14.08|
> > > |Low-entropy,$\gamma=4.0$|0.980/1.000|1.000/1.000|16.72/19.13|
> > >
> > > We then evaluate robustness under BIRA using the 50th-percentile threshold, which gives the better tradeoff. As shown below, this variant retains somewhat higher TPR after rewriting than the standard SIR, but the absolute TPR remains low. Intuitively, placing the signal on lower-entropy positions makes it less directly exposed to BIRA, but the usable signal becomes sparse. Paraphrasing still weakens it by changing both the surrounding context and the subset selected at detection time. In addition, rewriting often removes the grammatical errors and unusual repetitive phrases introduced by large $\gamma$, which further weakens the watermark signal used by the detector. The lower LLM-judge scores are also partly due to the poorer quality of the source watermarked text, which makes faithful paraphrasing harder. In manual inspection, texts with large $\gamma$ were often difficult to interpret because of the accumulated errors. Overall, robustness improves only modestly, while the text-quality cost is substantial.
> > >
> > > |Setting|TPR@FPR=1%|TPR@FPR=10%|PPL|LLM Judge|
> > > |-|-|-|-|-|
> > > |SIR|0.012|0.114|10.19|4.24|
> > > |Low-entropy,$\gamma=1.0$|0.096|0.230|10.50|4.26|
> > > |Low-entropy,$\gamma=2.0$|0.134|0.310|10.93|4.05|
> > > |Low-entropy,$\gamma=4.0$|0.206|0.382|11.44|3.86|
> > >
> > > > (2) Distributing the signal more evenly across positions.
> > >
> > > For (2), we strengthen the standard SIR with $\gamma=1.0$ by increasing $\gamma$, which spreads stronger watermark evidence across more positions, including lower-entropy ones. The table below shows that detectability remains high, but text quality degrades substantially as $\gamma$ increases, indicating stronger distortion of the model distribution and much worse PPL.
> > >
> > > |Setting|TPR@FPR=1%|TPR@FPR=10%|PPL|
> > > |-|-|-|-|
> > > |SIR,$\gamma=1.0$|0.998|1.000|11.95|
> > > |SIR,$\gamma=2.0$|1.000|1.000|15.70|
> > > |SIR,$\gamma=4.0$|1.000|1.000|19.16|
> > >
> > > The attack results below show that increasing $\gamma$ improves robustness somewhat, but the gain remains limited because the additional signal on lower-entropy positions contributes only modestly, as also seen in (1). The lower LLM-judge scores also reflect the poorer quality of the source watermarked text, which makes faithful rewriting harder. Therefore, distributing the signal more evenly does not provide a free defense and mainly trades off quality for moderate robustness gains.
> > >
> > > |Setting|TPR@FPR=1%|TPR@FPR=10%|PPL|LLM Judge|
> > > |-|-|-|-|-|
> > > |SIR,$\gamma=1.0$|0.012|0.114|10.19|4.24|
> > > |SIR,$\gamma=2.0$|0.032|0.138|10.42|3.96|
> > > |SIR,$\gamma=4.0$|0.064|0.210|10.71|3.72|
> > >
> > >
> > > Overall, for both (1) and (2), these variants make BIRA somewhat less effective only at a clear practical cost, requiring stronger watermark bias and noticeably degrading text quality. We will include these results in the manuscript to further strengthen the paper.

---

### Official Review · Reviewer_xDcT · 2026-03-12

**Soundness:** 2
**Presentation:** 3
**Significance:** 3
**Originality:** 3
**Overall Recommendation:** 4
**Confidence:** 5

**Summary:**

The paper proposes a query-free black-box rewriting attack for LLM watermarks based on a novel way to estimate the approximate greenlist (quantile cutoff based on proxy model surprisal). The attack achieves high evasion rates while preserving text quality and works on a broad set of schemes.

**Compliance With Llm Reviewing Policy:**

Affirmed.

**Final Justification:**

The authors have addressed all my concerns in the rebuttal so I raise my score from 3 to 4.

**Key Questions For Authors:**

- I like the explanations of transferability beyond token-level red-green schemes (this point could also be better emphasized in introduction), but I did not find any explanations that relate to EXP, and no mention of Synthid-text which is an interesting case. Could the authors coment on this?
- Q: On the adaptive bias, did the authors explore the idea of making the suffix bias-heavy instead of the prefix when presumably the key content of the text was already preserved? This feels like it could better preserve quality. On a related note, did the authors think how would localized detectors interact with this kind of non-uniformity in green tokens after the attack. Perhaps WinMax (from the original KGW paper) would do better as the suffix remains more green?
- Why is T5 cited for watermark attacks in the introduction? I find this quite unusual.

**Limitations:**

I did not find a limitations section; it would be good to include one. I don't think the societal impact section is necessary here.

**Strengths And Weaknesses:**

The paper is thorough and detailed and mostly well-positioned w.r.t. prior work on the important question of watermark robustnses (paraphrasers do exist in many forms but the query-free framing makes sense). The experiments are generally well executed and take into account quality evaluation and choose metrics carefully. The theory is a welcome addition but in my opinoin not the strongest point of the paper as the result is somewhat unsurprising for those familiar with prior literature on removal attacks. The paper is well written and contains many interesting additions such as the ablation studies and the discussion of the adaptive bias.

However, I have two key concerns with the evaluation:\
(1) The watermarked model seems to always be OPT 1.3B which as unusually low capability model, esp. compared to paraphrasers which are 8B+ or gpt4o. I am worried that by using only paraphrasers that are several capability levels above the watermarked model, the removal quality may be partially attributed to better text understanding. In practice the expectation is that the relative capability would be opposite: a weaker proxy model trying to remove a watermark from a SOTA proprietary LLM. \
(2) All text is very short (230 tokens per prompt) -- as discussed in Kirchenbauer's paper on reliability of KGW and many followups, it is known that removal is possible and easy on shorter texts. This is also evidenced by the results on some schemes in this submission being already very high for baselines. Experiments on longer texts (500/600+) tokens would be much mroe convincing.

I understand it may not be possible for authors to make such big changes to evaluation in a short time but I the two points above leave me unconvinced that the practical utility of BIRA translates beyond the specific setup chosen by the authors. I would be happy to recommend the paper for acceptance should more evidence be provided.

Some other relatively minor writing comments:
- On "L22 right" the authors mention that prior work "assumes unrestricted access to the victim model". The most reasonable understanding of this is "access to weights" which is not true as the cited work only needs query access. This should be clarified.
- On a similar note "Limited practicality in rate-limited API settings" strikes me as too harsh and misses the point that the citeed attack needs 10k queries once, after which attacks can be mounted for free. To not mislead readers, I would clarify this point and focus on the two stronger differentiating points (knowing the scheme and no need for top-k access).
- Still around the same spot, "graybox setting" seems to refer to "knowing parts of the scheme". The default understing of graybox is generally w.r.t. model access (so e.g. top-k logits) so this is again slightly misleading. I advise the authors to clearly distinguish model-access threat model and scheme-knowledge threat model; the underyling setup is convincing but the wording does not do it justice.
- Nit: Table 1 is not convincing without the context of Figure 2 (quality eval); I suggest the authors prefetch the quality analysis into Table 1 such that readers skimming do not leave with the impression that on some schemes the baselines are comparable to BIRA (when in reality they have much worse quality).

---

> ### Author Rebuttal · Authors · 2026-03-30
>
> We sincerely appreciate your taking the time to review and engage with our paper. We have carefully considered your thoughtful comments and provide detailed responses below.
>
> > W. Watermark generation with a strong model and longer text
>
> Our setup follows the state-of-the-art baseline SIRA to ensure a fair comparison. To further strengthen the evaluation, we additionally test BIRA in a longer-generation setting, using Qwen2.5-32B to generate 500 high-quality, 600-token watermarked texts and Llama-3.1-8B as the rewriting model.  Due to time and space constraints, we report results only for SIR, the most robust watermark in our experiments. The table below shows that BIRA remains highly effective in this setting, achieving a substantially lower TPR at a fixed FPR while maintaining a high LLM-judge score. By contrast, SIRA exhibits severe quality degradation, because masking becomes increasingly disruptive as text length grows. We will include these results in the revised manuscript.
>
> |Attack|TPR@FPR=1%↓|TPR@FPR=10%↓|Best F1↓|LLM Judge↑|PPL↓|Self-BLEU↓|
> |-|-|-|-|-|-|-|
> |Vanilla|57.7%|81.8%|86.8%|4.65|7.13|0.24|
> |DIPPER-1|41.2%|78.6%|86.5%|3.70|8.50|0.25|
> |DIPPER-2|25.2%|60.2%|82.3%|3.25|8.90|0.19|
> |SIRA|20.8%|48.4%|73.8%|1.33|7.19|0.02|
> |BIRA (Ours)|4.0%|11.0%|29.4%|4.14|8.90|0.05|
>
> > Comment: Wording (“Unrestricted access”, “gray-box”, “Limited practicality in rate-limited API settings”)
>
> Our intended meaning of unrestricted access is that the adversary can interact with the watermarked LLM through queries, rather than having access to model weights. This distinction is practically relevant because a user may wish to remove a watermark from text of unknown provenance without knowing which model or watermarking scheme produced it. We will revise the wording to make this clearer.
>
> For the gray-box setting, we meant knowledge of the watermarking scheme rather than access to the underlying model, and we will revise this wording to avoid misleading the reader.
>
> Regarding practicality in rate-limited API settings, we agree that the attack cost in prior work is relatively modest (10k queries or 50$ for GPT), although it may not be strictly one-time in practice if watermark keys are updated regularly for security. We will therefore revise the manuscript to present this point more carefully and instead emphasize the two stronger advantages of our method: it requires neither interaction with the watermarked LLM nor knowledge of the watermarking scheme.
>
> > Nit: Table 1 is not convincing without the context of Figure 2
>
> We thank the reviewer for this helpful suggestion. We will revise the manuscript to move the quality analysis into Table 1 to better highlight our results.
>
>
> > Q1. Transferability to EXP and SynthID
>
> Transferability to EXP and SynthID arises because both methods embed watermark traces more effectively in high-entropy settings, and detection relies on those traces. BIRA weakens this by suppressing token choices that are more likely to embed such traces, using self-information as a proxy. In EXP, detectability scales with the watermark potential defined in the EXP paper, which is larger when sampled tokens have lower conditional probability under the model. In SynthID, tournament sampling preferentially selects tokens with higher scores under the watermarking functions, and the paper notes that detectability is stronger when the LLM distribution has higher entropy. Hence, although EXP and SynthID differ algorithmically, BIRA consistently reduces detectability by suppressing the token choices that most strongly embed watermark evidence.
> We will add this explanation to the paper, along with additional results on SynthID. In our experiments under the setup in Section 5.1, BIRA reduces TPR@FPR=1% to 1.0% on SynthID.
>
> > Q2. Suffix bias-heavy and localized detector WinMax
>
> A suffix-heavy bias schedule is an interesting idea. However, in our setting, rewriting may reorder content or merge sentences, making it difficult to define a reliable suffix region on which to apply stronger bias. We therefore use a global proxy-based suppression mechanism, rather than position-dependent biasing, which remains effective under such structural changes and suppresses the watermark signal broadly instead of concentrating residual traces in a particular span. Consistent with this, BIRA also remains effective against the localized WinMax detector: for 500 KGW watermarked texts generated by Qwen2.5-32B with 600-token outputs and rewritten by Llama-3.1-8B, BIRA achieves 5.0% TPR at 1.0% FPR. Nevertheless, developing improved bias strategies that further optimize text quality is a promising direction for future work, and we will add this discussion and result to the manuscript.
>
>
> > Q3. T5 citation.
>
> We intended to cite the DIPPER baseline, which uses a T5-based model for paraphrase generation. We will correct this in the revision.

---

> > ### Author Rebuttal · Reviewer_xDcT · 2026-04-04
> >
> > Thank you for a well written rebuttal. The answer to Q1 is especially interesting and I appreciate the new experiment. I raise my score as my concerns were sufficiently adressed.

---

> > > ### Author Response · Authors · 2026-04-05
> > >
> > > We sincerely appreciate your constructive feedback, which has been very helpful in strengthening the paper. We will incorporate these additional results and discussion into the revised manuscript. We also greatly appreciate your positive reassessment and for raising the score.

---

### Official Review · Reviewer_bgKn · 2026-03-13

**Soundness:** 3
**Presentation:** 3
**Significance:** 3
**Originality:** 3
**Overall Recommendation:** 4
**Confidence:** 3

**Summary:**

This paper studies a very practical and important question in LLM watermarking, namely whether a black-box attacker can remove watermark signals by simple rewriting without badly hurting meaning. The main idea is clean and easy to follow. The theoretical analysis gives a nice intuition that one only needs to consistently reduce the average probability of sampling green tokens, and the proposed BIRA attack turns that insight into a simple decoding-time intervention with negative logit bias. Empirically, the attack is very strong across several watermarking schemes and language models, and the semantic fidelity results are generally better than prior rewriting baselines. Overall, I think the paper is timely, well motivated, and surprisingly effective for such a lightweight method.

**Compliance With Llm Reviewing Policy:**

Affirmed.

**Final Justification:**

Sorry for the misplacement of some of my reviews. My evaluation score of the work is originally correct. Hoping to discuss more with the authors.

**Key Questions For Authors:**

See weakness.

**Limitations:**

See weakness.

**Strengths And Weaknesses:**

## Strengths
1. The paper has a very clear story from theory to method to experiments, and the main idea is simple enough that people will actually remember it.

2. The empirical attack results are strong and quite convincing, especially the near-saturated ASR across many watermark variants.

3. I also like that the method is training-free and black-box on the victim side, which makes the threat model much more realistic than many prior attack papers.




## Weaknesses
1. The theory really talks about the true green set, but the method only uses a surprisal-based proxy. It works in the experiments, but the paper still doesn’t fully show when this proxy is actually reliable and when it may just miss the real watermark trace.

2. BIRA kind of depends on having token-level logit bias at decoding time. That is fine for some open models or a few APIs, but not a very universal setting, so the practical attack story feels a bit narrower than it first sounds.

3. The eval setup is still a bit too fixed. Since surprisal is very sensitive to temperature and decoding config, more tests there can strengthen the experiments. Right now it’s hard to tell how stable the method is once those settings shift.

---

### Official Review · Reviewer_kQwM · 2026-03-13

**Soundness:** 3
**Presentation:** 4
**Significance:** 3
**Originality:** 2
**Overall Recommendation:** 4
**Confidence:** 3

**Summary:**

This paper designs a new evasion attack for watermarked text. In essence, the paper gives the BIRA algorithm which will resample some low entropy tokens in a watermarked text, after putting some negative biais on the logits associated with these tokens.
The paper also presents a theorem showing that the detection probability tends exponentially quickly to zero when the sampling probabilities of green tokens are altered. Experiments show that the BIRA algorithm achieves state of the art evasion attack.

**Compliance With Llm Reviewing Policy:**

Affirmed.

**Key Questions For Authors:**

- When BIRA samples new replacement tokens, why does it use M(\hat{y}^0:N-1) instead of M(\tilde{y}^0:N-1) ? shouldn't we feed the sampling LLM with the new string instead of the original one ?
- the heuristic in BIRA relies solely on not replacing important tokens (those with high self-information). Wouldn't it be possible to refine this heuristic to by estimating how likely it is that the current original token was in the green set ?
- currently, BIRA creates a "proxy suppression set" of tokens which suppression might have a low impact on the meaning of the text. But assume a string mixes python code and normal text. Then some tokens will have high self information in the code but low information in the natural language text. Wouldn't it be better to handle this kind of situation ?

**Limitations:**

yes

**Strengths And Weaknesses:**

- Overall, the paper is sound. The math is correct, eventhough not groundbreaking: Lemma 4.1 is trivial (and should probably not be included in the main text), and Theorem 4.2 is a stochastic version of the following trivial claim "if the attacker manages to alter the string y so as to be below the detection threshold, then it will not be detected".

- The BIRA algorithm is also sound. It mainly attempts to resample tokens whose replacement will be of low impact (because they are of low-self information). This seems a very nice proxy heuristic, although there might be better heuristics if the goal is to maintain semantic similarity with the original string.

- The paper is *very* well written, easy and very pleasant to read. The structure is nice, notations are well chosen.

- This topic (watermarking and evasion attacks) is of high social impact currently, and i would say the paper brings a small but interesting contribution to the field (although i am not an expert here).

- The experimental part of the paper is globally good. They are quite extensive (many datasets, many watermarking schemes,...). Results are very encouraging and show that BIRA outperforms previous methods. However, the setting might not be highly realistic (the strings which are watermarked are very short)

---

> ### Author Rebuttal · Authors · 2026-03-28
>
> We sincerely appreciate your taking the time to review and engage with our paper. We have carefully considered your thoughtful comments and provide detailed responses below.
>
> > Contribution of our theorem
>
> Its main role is to formalize rewriting-based watermark evasion in an analyzable framework and to provide theoretical justification for the design of BIRA. Specifically, the result shows that even a small reduction in the probability of sampling green tokens during rewriting can suffice for successful evasion. This directly motivates our bias-inversion strategy, which applies a negative logit bias to a proxy suppression set during decoding. We appreciate the reviewer’s suggestion that Lemma 4.1 could be moved to the appendix, although we currently retain it in the main text for readability.
>
> > The setting might not be highly realistic (short length of the watermarked text)
>
> Our setup follows the state-of-the-art baseline SIRA to ensure a fair comparison. To further strengthen the evaluation, we additionally evaluate BIRA in a longer-generation setting, using Qwen2.5-32B to generate 500 high-quality, 600-token texts watermarked with SIR, the most robust watermark in our experiments, and Llama-3.1-8B as the rewriting model. As shown in the table below, BIRA remains highly effective in this setting, achieving a substantially lower TPR at a fixed FPR while maintaining a high LLM-judge score. In contrast, SIRA suffers severe quality degradation, as masking becomes increasingly disruptive for longer texts. We will include these results in the revised manuscript.
>
> |Attack|TPR@FPR=1%↓|TPR@FPR=10%↓|Best F1↓|LLM Judge↑|PPL↓|Self-BLEU↓|
> |-|-|-|-|-|-|-|
> |Vanilla|57.7%|81.8%|86.8%|4.65|7.13|0.24|
> |DIPPER-1|41.2%|78.6%|86.5%|3.70|8.50|0.25|
> |DIPPER-2|25.2%|60.2%|82.3%|3.25|8.90|0.19|
> |SIRA|20.8%|48.4%|73.8%|1.33|7.19|0.02|
> |BIRA (Ours)|4.0%|11.0%|29.4%|4.14|8.90|0.05|
>
> > When BIRA samples new replacement tokens, why does it use $M(\hat{y}^{0:N-1})$ instead of $M(\tilde{y}^{0:N-1})$ ?
> shouldn't we feed the sampling LLM with the new string instead of the original one ?
>
> We believe that the reviewer is referring to Line 14 of Algorithm 1. In our formulation, the rewrite model $M$ is conditioned on three components: the current rewritten prefix $\tilde{y}$, the original watermarked source text $\hat{y}^{0:N-1}$, and the system prompt $S$ for rewriting, denoted as $M(\tilde{y}, \hat{y}^{0:N-1}, S)$. This ensures that each new token is generated based on the updated partial rewrite while using the original source text as guidance for the rewriting process.
>
>
> > the heuristic in BIRA relies solely on not replacing important tokens (those with high self-information). Wouldn't it be possible to refine this heuristic to by estimating how likely it is that the current original token was in the green set ?
>
> In the black-box setting, the true green set is secret and inaccessible, so directly estimating whether a token belongs to the green set would generally require additional assumptions or query access beyond our threat model. Instead, we use self-information as a simple and effective proxy for constructing the suppression set, since watermarking is most effective in high-entropy contexts. Exploring more refined proxies under the same black-box constraint is an interesting direction for future work.
>
> > currently, BIRA creates a "proxy suppression set" of tokens which suppression might have a low impact on the meaning of the text. But assume a string mixes python code and normal text. Then some tokens will have high self information in the code but low information in the natural language text. Wouldn't it be better to handle this kind of situation ?
>
> Thank you for the insightful suggestion. Exploring mixed code-and-natural-language inputs is an interesting and valuable direction. BIRA could be further improved in such cases through domain-aware proxy-construction strategies. For instance, one possible extension would be to first separate code spans from natural-language spans, apply bias inversion to each span independently, and then recombine the rewritten spans into a single output.
>
> Additionally, BIRA is less brittle here because it applies a soft negative bias rather than banning tokens outright. This reduces pressure to suppress tokens that are important for faithful rewriting, while actual token generation remains context-dependent. As a result, tokens that are strongly favored by the model can still be produced when necessary. This is also consistent with our empirical results showing that BIRA achieves high semantic fidelity. Nevertheless, we acknowledge that systematically exploring such cases and alternative strategies remains a promising direction for further improving text quality.

---

> > ### Author Rebuttal · Reviewer_kQwM · 2026-04-05
> >
> > thank you to the authors for their answers.

---

> > > ### Author Response · Authors · 2026-04-05
> > >
> > > We appreciate the careful review and insightful comments, which have helped strengthen our paper. We will incorporate these into the final manuscript.

---

### Decision · Program_Chairs · 2026-04-30

**Decision:**

Accept (regular)

**Comment:**

This paper proposes BIRA, a query-free black-box evasion method against LLM watermarking that applies negative logit bias to a proxy suppression set identified via token surprisal, supported by a theoretical analysis showing that even a small reduction in green token sampling probability causes detection probability to decay exponentially. Prior to the rebuttal, reviewers broadly acknowledged the paper's clean theory-to-method narrative, strong empirical performance, and realistic threat model, but raised shared concerns about the heuristic nature of the proxy set, limited evaluation on robust watermarking schemes, and the reliance on cross-model tokenizer alignment; notably, one review (Reviewer bgKn) was identified as misplaced and intended for a different paper, and was accordingly down-weighted by the AC. After the rebuttal and discussion, Reviewers kQwM and xDcT fully resolved their concerns and maintained or raised their scores, while Reviewer h4Ch acknowledged that most issues were addressed but maintained the score, noting that empirical analysis of failure modes under shifted watermark signals would further strengthen the paper's claims. By trading off the strengths and weaknesses of this paper, as well as the reviewers' largely positive post-rebuttal assessments, I decide to accept this paper.